# Matrix Encoding Networks
# for Neural Combinatorial Optimization

**Yeong-Dae Kwon, Jinho Choo, Iljoo Yoon, Minah Park, Duwon Park, Youngjune Gwon**
Samsung SDS
{y.d.kwon, jinho12.choo, iljoo.yoon, minah86.park, dw21.park, gyj.gwon}@samsung.com

## Abstract

Machine Learning (ML) can help solve combinatorial optimization (CO) problems better. A popular approach is to use a neural net to compute on the parameters of a given CO problem and extract useful information that guides the search for good solutions. Many CO problems of practical importance can be specified in a matrix form of parameters quantifying the relationship between two groups of items. There is currently no neural net model, however, that takes in such matrix-style relationship data as an input. Consequently, these types of CO problems have been out of reach for ML engineers. In this paper, we introduce Matrix Encoding Network (MatNet) and show how conveniently it takes in and processes parameters of such complex CO problems. Using an end-to-end model based on MatNet, we solve asymmetric traveling salesman (ATSP) and flexible flow shop (FFSP) problems as the earliest neural approach. In particular, for a class of FFSP we have tested MatNet on, we demonstrate a far superior empirical performance to any methods (neural or not) known to date.

## 1 Introduction

Many combinatorial optimization (CO) problems of industrial importance are NP-hard, leaving them intractable to solve optimally at a large scale. Fortunately, researchers in operations research (OR) have developed ways to tackle these NP-hard problems in practice, mixed integer programming (MIP) and meta-heuristics being two of the most general and popular approaches. With the rapid progress in deep learning techniques over the last several years, a new approach based on machine learning (ML) has emerged. ML has been applied in both ways successfully [1], as a helper for the traditional OR methods aforementioned or as an independent CO problem solver trained in an end-to-end fashion.

One way to leverage ML for solving CO problems is to employ a "front-end" neural net, which directly takes in the data specifying each problem. The neural net plays a critical role of analyzing all input data as a whole, from which it extracts useful global information. In an end-to-end ML-based approach, such global information may be encoded onto the representations of the entities making up the problem. Greedy selection strategies based on these representations are then no longer too near-sighted, allowing globally (near-) optimal solutions to be found quickly. Existing OR methods can also benefit from incorporating the global information. Many hybrid approaches already exist that leverage information extracted by a neural net, both in ML-MIP forms [2, 3] and ML-heuritic forms [4, 5, 6].

The literature on neural combinatorial optimization contains many different types of the front-end models, conforming to the variety of data types upon which CO problems are defined. Yet, there has been no research for a model that encodes matrix-type data. This puts a serious limitation on the range of CO problems that an ML engineer can engage. Take, for example, the traveling salesman problem (TSP), the most intensely studied topic by the research community of neural combinatorial

35th Conference on Neural Information Processing Systems (NeurIPS 2021).

optimization. Once we lift the Euclidean distance restriction (the very first step towards making the problem more realistic), the problem instantly becomes a formidable challenge that has never been tackled before because it requires a neural net that can process the distance matrix.[1]

The list of other classical CO problems based on data matrices includes job-shop/flow-shop scheduling problems and linear/quadratic assignment problems, just to name a few. All of these examples are fundamental CO problems with critical industrial applications. To put it in general terms, imagine a CO problem made up of two different classes of items, $\tilde{A} = \{\tilde{a}_1, \ldots, \tilde{a}_M\}$ and $\tilde{B} = \{\tilde{b}_1, \ldots, \tilde{b}_N\}$, where $M$ and $N$ are the sizes of $\tilde{A}$ and $\tilde{B}$ respectively. We are interested in the type where features of each item are defined by its relationships with those in the other class. The data matrix we want to encode, $\mathbf{D} \in \mathbb{R}^{M \times N}$, would be given by the problem,[2] where its $(i, j)$th element represents a quantitative relationship of some sort between a pair of items $(\tilde{a}_i, \tilde{b}_j)$. The sets $\tilde{A}$ and $\tilde{B}$ are unordered lists, meaning that the orderings of the rows and the columns of $\mathbf{D}$ are arbitrary. Such permutation invariance built into our data matrices is what sets them apart from other types of matrices representing stacked vector-lists or a 2D image.

In this paper, we propose Matrix Encoding Network (MatNet) that computes good representations for all items in $\tilde{A}$ and $\tilde{B}$ within which the matrix $\mathbf{D}$ containing the relationship information is encoded. To demonstrate its performance, we have implemented MatNet as the front-end models for two end-to-end reinforcement learning (RL) algorithms that solve the asymmetric traveling salesman (ATSP) and the flexible flow shop (FFSP) problems. These classical CO problems have not been solved using deep neural networks. From our experiments, we have confirmed that MatNet achieves near-optimal solutions. Especially, for the specific FFSP instances we have investigated, our MatNet-based approach performs substantially better than the conventional methods used in operations research including mixed integer programming and meta-heuristics.

## 2 Related work

A neural net can accommodate variable-length inputs as is needed for encoding the parameters of a CO problem. Vinyals *et al.* [9], one of the earliest neural CO approaches, have introduced Ptr-Nets that use a recurrent neural network (RNN) [10] as the front-end model. Points on the plane are arranged into a sequence, and RNN processes the Cartesian coordinates of each point one by one. Bello *et al.* [11] continue the RNN approach, enhancing the Ptr-Net method through combination with RL. Nazari *et al.* [12] further improve this approach by discarding the RNN encoder, but keeping the RNN decoder to handle the variable input size.

The graph neural network (GNN) [13, 14] is another important class of encoding networks used for neural CO, particularly on (but not limited to) graph problems. GNN learns the message passing policy between nodes that can be applied to an arbitrary graph with any number of nodes. Khalil *et al.* [7] are among the firsts to show that a GNN-based framework can solve many different types of CO problems on graph in a uniform fashion. Li *et al.* [15] extend the work by switching to a more advanced structure of graph convolutional networks (GCNs) [16]. Manchanda *et al.* [17] demonstrate efficient use of GCN that can solve CO problems on graphs with billions of edges. Karalias and Loukas [18] use unsupervised learning on GNN to improve the quality of the CO solutions.

Encoding networks most relevant to our work are those based on the encoder of the Transformer [19]. Note that this encoder structure can be viewed as a graph attention network (GAT) [20] operating on fully connected graphs. Using this type of encoder, Kool *et al.* [21] encode Cartesian coordinates of all nodes given by the problem simultaneously and solve the TSP and many other related routing problems. Kwon *et al.* [8] use the same encoder as Kool *et al.* but produce solutions of significantly improved quality with the use of the RL training and the inference algorithms that are more suitable for combinatorial optimizations.

---

[1]One could, in principle, consider a general GNN approach such as Khalil *et al.* [7] that can encode any arbitrary graphs. However, as can be seen from the comparison of the TSP results of Khalil *et al.*( >8% error on random 100-city instances) with some of the state-of-the-art ML results (*e.g.*, Kwon *et al.* [8], <0.2% error), a general GNN is most suitable for problems that deal with diverse graph structures, not those with fixed, dense graph structures (such as matrices).

[2]The relationship can have multiple ($f$) features, in which case we are given $f$ number of different data matrices $\mathbf{D}^1$, $\mathbf{D}^2$, $\ldots$, $\mathbf{D}^f$. For simplicity we only use $f$=1 in this paper, but our model covers $f$>1 cases as well. (See Appendix A.1).

MatNet architecture is designed to encode a bipartite graph, as will be explained in detail in the following section. Although none support embedding of a single matrix like MatNet, many interesting deep learning works exist that rely on bipartite graph embeddings or matrix embeddings. Gasse *et al.* [22] learn effective branch-and-bound variable selection policies for solving MIP instances, where constraints and variables of an MIP instance form a bipartite graph weighted by constraint coefficients. Gibbons *et al.* [23] solve weapon-target assignment problem, a classic CO problem that is presented with a bipartite graph. Duetting *et al.* [24] find good auction policies given a matrix that describes the value that each bidder assigns to each auctioning item.

# 3 Model architecture

MatNet can be considered a particular type of GNN that operates on a complete bipartite graph with weighted edges. The graph has two sets of nodes, $A = \{a_1, \ldots, a_M\}$ and $B = \{b_1, \ldots, b_N\}$, and the edge between $a_i$ and $b_j$ has the weight $e(a_i, b_j) \equiv \mathbf{D}_{ij}$, the $(i, j)$-th element of $\mathbf{D}$, as illustrated in Figure 1.

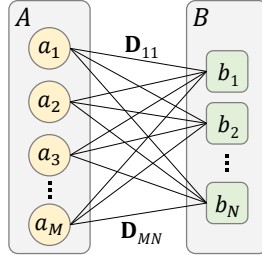

MatNet is inspired by the encoder architecture of the Attention Model (AM) by Kool *et al.* [21], but because the AM deals with a quite different type of a graph (fully connected nodes and no edge weights), we have made significant modifications to it. The encoder of the AM follows the node-embedding framework of graph attention networks (GATs) [20], where a model is constructed by stacking multiple ($L$) graph attentional layers. Within each layer, a node $v$'s vector representation $\hat{h}_v$ is updated to $\hat{h}'_v$ using aggregated representations of its neighbors as in

Figure 1: A complete bipartite graph with weighted edges.

$$\hat{h}'_v = \mathcal{F}\big(\hat{h}_v, \{\hat{h}_w \mid w \in \mathcal{N}_v\}\big). \tag{3.1}$$

Here, $\mathcal{N}_v$ is the set of all neighboring nodes of $v$. The (learnable) update function $\mathcal{F}$ is composed of multiple attention heads, and their aggregation process utilizes the attention score for each pair of nodes $(v, w)$, which is a function of $\hat{h}_v$ and $\hat{h}_w$.

MatNet also generally follows the GATs framework, but it differs in two major points: (1) It has two independent update functions that separately apply to nodes in $A$ and $B$. (2) The attention score for a pair of nodes $(a_i, b_j)$ is not just a function of $\hat{h}_{a_i}$ and $\hat{h}_{b_j}$, but the edge weight $e(a_i, b_j)$ as well. The update function in each layer of MatNet can be described as

$$
\begin{aligned}
\hat{h}'_{a_i} &= \mathcal{F}_A\big(\hat{h}_{a_i}, \{(\hat{h}_{b_j}, e(a_i, b_j)) \mid b_j \in B\}\big) \quad \text{for all } a_i \in A, \\
\hat{h}'_{b_j} &= \mathcal{F}_B\big(\hat{h}_{b_j}, \{(\hat{h}_{a_i}, e(a_i, b_j)) \mid a_i \in A\}\big) \quad \text{for all } b_j \in B.
\end{aligned}
\tag{3.2}
$$

## 3.1 Dual graph attentional layer

The use of two update functions $\mathcal{F}_A$ and $\mathcal{F}_B$ in Eq. (3.2) is somewhat unorthodox. The GATs framework requires that the update rule $\mathcal{F}$ be applied uniformly to all nodes because it loses its universality otherwise. In our case, however, we focus on a particular type of a problem, where the items in $\tilde{A}$ and $\tilde{B}$ belong to two qualitatively different classes. Having two separate functions allows MatNet to develop a customized representation strategy for items in each class.

The two update functions are visualized as a dual structure of a graph attentional layer depicted in Figure 2(a). The MatNet architecture is a stack of $L$ graph attentional layers, and each layer is made of two sub-blocks that implement $\mathcal{F}_A$ and $\mathcal{F}_B$. The two sub-blocks are structurally identical, where each one is similar to the encoder layer of the AM, which in turn is based on that of the Transformer model [19]. Note, however, that while both the AM and the Transformer model use the simple self-attention for encoding, MatNet requires a clear separation of queries (nodes in one set) and the key-value pairs (nodes in the other set) at the input stage to perform cross-attentions. For detailed descriptions of computing blocks used in Figure 2(a), we refer the readers to the Transformer architecture [19].

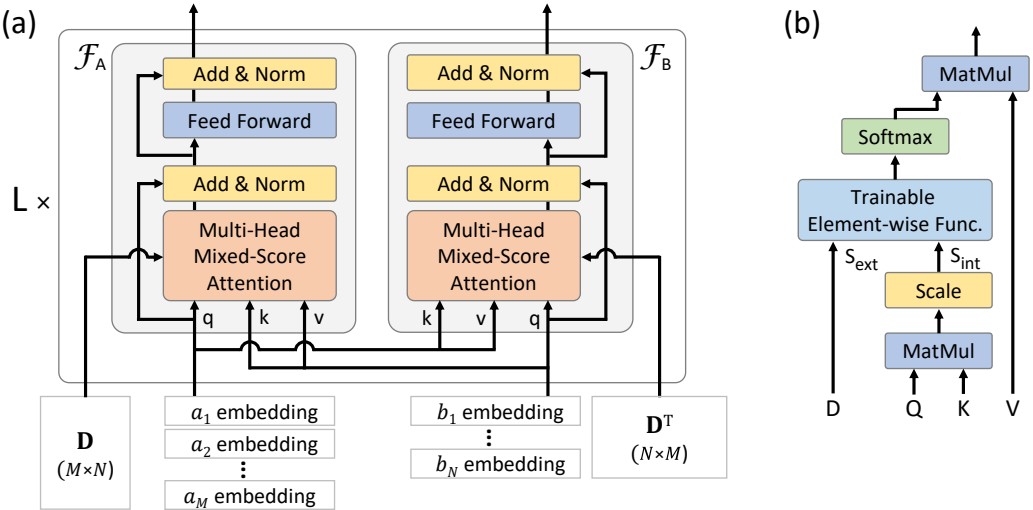

Figure 2: (a) An overview of the MatNet architecture. (b) Mixed-score attention. "Multi-Head Mixed-Score Attention" block in (a) consists of many independent copies of the mixed-score attentions arranged in parallel and fully connected (FC) layers at the input and output interfaces (not drawn).

## 3.2 Mixed-score attention

We now focus on how MatNet makes use of the edge weight $e(a_i, b_j) \equiv \mathbf{D}_{ij}$ in its attention mechanism. "Multi-Head Mixed-Score Attention" block in Figure 2(a) is the same as "Multi-Head Attention" block of the Transformer except that the scaled dot-product attention in each attention head is replaced by the mixed-score attention shown in Figure 2(b). Originally, the scaled dot-product attention outputs the weighted sum of the values, where the weights are calculated from the attention scores (the scaled dot-products) of query-key pairs. For MatNet, in addition to these internally generated scores, it must use the externally given relationship scores, $\mathbf{D}_{ij}$, which is also defined for all query-key pairs. The mixed-score attention mechanism mixes the (internal) attention score and the (external) relationship score before passing it to the next "SoftMax" stage. We note that mixing of the two scores appear in the works of Sykora *et al.* [25] and Dwivedi *et al.* [26] in similar ways.

Rather than using a handcrafted prioritizing rule to combine the two types of the scores, we let the neural net figure out the best mixing formula. "Trainable element-wise function" block in Figure 2(b) is implemented by a multilayer perceptron (MLP) with two input nodes and a single output node. Each head in the multi-head attention architecture is given its own MLP parameters, so that some of the heads can learn to focus on a specific type of the scores while others learn to balance, depending on the type of graph features that each of them handles. Note that it is an element-wise function, so that the output of the mixed-score attention is indifferent to the ordering of $a_i$ and $b_j$, ensuring the permutation-invariance of the operation.

The attention mechanism of the Transformer is known for its efficiency as it can be computed by highly optimized matrix multiplication codes. The newly inserted MLP can also be implemented using matrix multiplications without the codes that explicitly loop over each row and column of the matrix $\mathbf{D}$ or each attentional head. Actual implementation of MatNet keeps the efficient matrix forms of an attention mechanism. Also, note that the duality of the graph attentional layers of MatNet explained in Section 3.1 is conveniently supported by $\mathbf{D}^{\mathrm{T}}$, a transpose of the matrix $\mathbf{D}$.

## 3.3 The initial node representations

The final pieces of MatNet that we have not yet explained are the initial node representations that are fed to the model to set it in motion. One might be tempted to use the edge weights directly for these preliminary representations of nodes, but the vector representations are ordered lists while edges of a graph offer no particular order, making this choice inappropriate.

MatNet uses zero-vectors to embed nodes in $A$, and one-hot vectors for nodes in $B$ (or vice versa) initially. The use of the same zero-embeddings for all nodes of $A$ is allowed, as long as all nodes in $B$ are embedded differently, because they acquire unique representations after the first graph attentional layer. The zero-vector embedding enables MatNet to support variable-sized inputs. Regardless of how the number of the nodes in $A$ changes from a problem instance to another, MatNet can process them all in a uniform manner. In Appendix H, for example, we demonstrate solving FFSP instances containing 1,000 different jobs using MatNet that is trained on smaller instances.

The one-hot vector embedding scheme for nodes in $B$, on the other hand, provides a somewhat limited flexibility towards varying $N$, the number of columns in the input matrix. Before a MatNet model is trained, its user should prepare a pool of $N_{\max}$ different one-hot vectors. When a matrix $\mathbf{D}_\circ$ with $N_\circ$ ($\leq N_{\max}$) columns is given, $N_\circ$ different one-hot vectors are randomly drawn from the pool in sequence and used as the initial node representations for $B$-nodes.

When a good value for $N_{\max}$ is difficult to establish before the deployment, or when the application requires better generalizability without the strict limit on $N$, one can employ an alternative initial node representation scheme. One-hot vectors can be substituted by vectors filled with the random numbers that are drawn independently for each problem instance. This completely lift the restriction of $N_{\max}$, at a price which slightly worsens the model's performance. (See Appendix A.3.)

**Instance augmentation via different sets of embeddings.** The initial one-hot (or random) vector embedding scheme naturally accommodates the "instance augmentation" strategy proposed by Kwon *et al.* [8]. For each random sequence of one-hot vectors chosen to embed $B$-nodes, a MatNet model encodes the matrix $\mathbf{D}$ in a different way. (That is, the number of node representation sets for a given problem instance can be *augmented* from one to many.) One can take advantage of this property and easily create many dissimilar solutions simply by running the model repeatedly, each time with a new one-hot vector sequence. Among those multiple solutions, the best one is chosen. This way, a higher quality solution can be acquired at the cost of increased runtime. The diversity of solutions created by the instance augmentation technique are far more effective than those produced by the conventional multi-sampling technique. (See Appendix C.)

## 4  Asymmetric traveling salesman problem

**Problem definition.** In the traveling salesman problem (TSP), the goal is to find the permutation of the given $N$ cities so that the total distance traveled for a round trip is minimized. For each pair of "from" city $a_i$ and "to" city $b_j$, a distance $d(a_i, b_j)$ is given, which constitutes an $N$-by-$N$ distance matrix. To verify that Mat-Net can handle general cases, we focus on the asymmetric traveling salesman problem (ATSP). This problem does not have the restriction $d(a_i, b_j) = d(a_j, b_i)$ so that its distance matrix is asymmetric. We use "tmat"-class ATSP instances that have the triangle inequality and are commonly studied by the OR community [27]. (See Appendix B for detailed explanation of the tmat class.) We solve the ATSP of three different sizes, having $N = 20$, $50$, and $100$ number of cities.

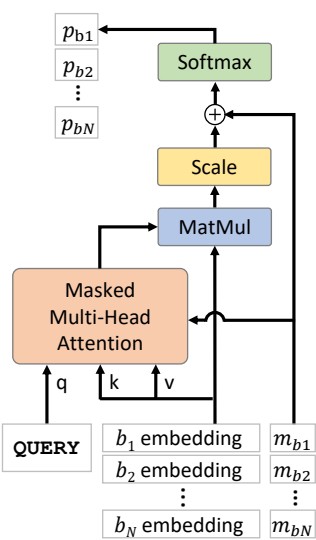

Figure 3: The decoder. Scalars $m_{bj}$ and $p_{bj}$ are the mask and the selection probability for $b_j$ ($j = 1, 2, \cdots, N$).

**MatNet configuration.** MatNet is used to encode the distance matrix. Regardless of the size of the ATSP, we have used the same MatNet structure. The MatNet model is constructed by stacking $L = 5$ encoding layers. The embedding dimension, $d_{\text{model}}$, for input/output of each layer is set to 256. Both sub-modules ($\mathcal{F}_A$ and $\mathcal{F}_B$) use $h = 16$ attention heads, where each head processes query, key, and value as 16-dimensional vectors (*i.e.*, $d_q = d_k = d_v = 16$). The score-mixing MLP in each head has one hidden layer with 16 nodes. For implementation of the "Scale" block in Figure 2(b), we apply scaling of $1/\sqrt{d_k}$ and then soft-clipping to $[-10, 10]$ using a $\tanh$ function, following the convention of Bello *et al.* [11]. "Feed-forward" blocks in Figure 2(a) is imple-

Table 4.1: Experiment results on 10,000 instances of ATSP

| Method | ATSP20 | | | ATSP50 | | | ATSP100 | | |
|---|---|---|---|---|---|---|---|---|---|
| | Len. | Gap | Time | Len. | Gap | Time | Len. | Gap | Time |
| CPLEX | 1.54 | - | (12m) | 1.56 | - | (1h) | 1.57 | - | (5h) |
| Nearest Neighbor | 2.01 | 30.39% | (-) | 2.10 | 34.61% | (-) | 2.14 | 36.10% | (-) |
| Nearest Insertion | 1.80 | 16.56% | (-) | 1.95 | 25.16% | (-) | 2.05 | 30.79% | (-) |
| Furthest Insertion | 1.71 | 11.23% | (-) | 1.84 | 18.22% | (-) | 1.94 | 23.37% | (-) |
| LKH3 | 1.54 | 0.00% | (1s) | 1.56 | 0.00% | (11s) | 1.57 | 0.00% | (1m) |
| MatNet | 1.55 | 0.53% | (2s) | 1.58 | 1.34% | (8s) | 1.62 | 3.24% | (34s) |
| MatNet ($\times$128) | 1.54 | 0.01% | (4m) | 1.56 | 0.11% | (17m) | 1.59 | 0.93% | (1h) |

mented with one hidden layer of dimension $d_{\text{ff}} = 516$. We use instance normalization for "Add & Norm." For the initial node representations, we use zero vectors for "from" cities and one-hot vectors to embed "to" cities. The size of the pool from which we draw the one-hot embedding vectors is adjusted to the minimum, the number of the cities of the problem (*i.e.*, $N_{\text{max}} = N$).

**Decoder.** Once MatNet processes the distance matrix $\mathbf{D}$ and produces vector representations of "from" and "to" cities, a solution of ATSP (a permutation sequence of all cities) can be constructed autoregressively, one city at a time. The decoder model of Kool *et al.* [21] (shown in Figure 3) is used repeatedly at each step. Two "from" city representations, one for the current city and the other for the first city of the tour, are concatenated to make the QUERY token.[3] Vector representations of all "to" cities go into the decoder, and for each city the decoder calculates the probability to select it as the next destination. A mask (0 for the unvisited and $-\infty$ for the visited cities) is used inside the multi-head attention block to guide the decision process, as well as right before the final softmax operation to enforce one-visit-per-city rule of the problem. See Appendix J for a diagram that explains the ATSP decoding process.

**Training.** We use the POMO training algorithm [8] to perform reinforcement learning on the MatNet model and the decoder. That is, for an ATSP with size $N$, $N$ different solutions (tours) are generated, each of them starting from a different city. The averaged tour length of these $N$ solutions is used as a baseline for REINFORCE algorithm [28]. We use Adam optimizer [29] with a learning rate of $4 \times 10^{-4}$ without a decay and a batch size of 200. With an epoch being defined as training 10,000 randomly generated problem instances, we train 2,000, 8,000, and 12,000 epochs for $N = 20$, 50, and 100, respectively. For $N = 20$ and 50, they take roughly 6 and 55 hours, respectively, on a single GPU (Nvidia V100). For $N = 100$, we accumulate gradients from 4 GPUs (each handling a batch of size 50) and the training takes about 110 hours. Multi-GPU processes are used for speed, as well as to overcome our GPU memory constraint (32GB each).[4]

**Inference.** We use the POMO inference algorithm [8]. That is, we allow the decoder to produce $N$ solutions in parallel, each starting from a different city, and choose the best solution. We use sampled trajectories for a POMO inference rather than greedy ones (argmax on action probabilities) [21] because they perform slightly better in our models.

## 4.1 Comparison with other baselines

In Table 4.1, we compare the performance of our trained models with those of other representative baseline algorithms on 10,000 test instances of the tmat-class ATSP. The table shows the average length of the tours generated by each method (displayed in units of $10^6$). The gap percentage is with respect to the CPLEX results. Times are accumulated for the computational processes only, excluding the program and the data (matrices) loading times. This makes the comparisons between

---

[3]Information of the first city is required in the QUERY token, because the decoder needs to know where the final destination of the tour is. The average of all node embeddings (a.k.a. "the graph embedding") is dropped from the QUERY token. See Appendix E for a discussion on this omission.)

[4]If needed, a smaller batch size can be used to reduce GPU memory usage. A batch size of 50 achieves the similar results when trained with a learning rate of $1 \times 10^{-4}$, although this takes a bit longer to converge.

different algorithms more transparent. Note that the heuristic baseline algorithms require only a single thread to run. Because it is easy to make them run in parallel on modern multi-core processors, we record their runtimes divided by 8. (They should be taken as references only and not too seriously.) For consistency, inference times of all MatNet-based models are measured on a single GPU, even though some models are trained by multiple GPUs.

Detailed explanations on all the baseline algorithms for the ATSP and the FFSP experiments, including how we have implemented them, are given in Appendix B and G, respectively.

**Mixed-integer programming (MIP).** Many CO problems in the industry are solved by MIP because it can model a wide range of CO problems, and many powerful MIP solvers are readily available. If one can write down a mathematical model that accurately describes the CO problem at hand, the software can find a good solution using highly engineered branch-and-bound type algorithms. This is done in an automatic manner, freeing the user from the burden of programming. Another merit of the MIP approach is that it provides an optimality guarantee. MIP solvers keep track of the gap between the best solution found so far and the best lower (or upper) bound for the optimal.

To solve test instances using MIP, we use one of the well-known polynomial MIP formulations of the ATSP [30] and let CPLEX [31] solve this model with the distance matrices that we provide. CPLEX is one of the popular commercial optimization software used by the OR community. With 10-second timeouts, CPLEX exactly solves all $N = 20$ instances. For $N = 50$ and 100 test instances, it solves about 99% and 96% of them to the optimal values, respectively. Solutions that are not proven optimal have the the average optimality gap of 1.2% for $N = 50$ and 0.5% for $N = 100$.

**Heuristics.** Nearest Neighbor (NN), Nearest Insertion (NI), and Furthest Insertion (FI) are simple greedy-selection algorithms commonly cited as baselines for TSP algorithms. Our implementations of NN, NI, and FI for ATSP in C++ solve 10,000 test instances very quickly, taking at most a few seconds (when $N = 100$). We thus omit their runtimes in Table 4.1.

The other heuristic baseline, LKH3 [32], is a state-of-the-art algorithm carefully engineered for various classical routing problems. It relies on a local search algorithm using $k$-opt operations to iteratively improve its solution. Even though it offers no optimality guarantee, our experiment shows that it finds the optimal solutions for most of our test instances in a very short time.

**MatNet.** In Table 4.1, the performance of the MatNet-based ATSP solver is evaluated by two inference methods: 1) a single POMO rollout, and 2) choosing the best out of 128 solutions generated by the instance augmentation technique (*i.e.*, random one-hot vector assignments for initial node embeddings) for each problem instance. The single-rollout inference method produces a solution in $N$ selection steps just like the other greedy-selection algorithms (NN, NI, and FI), but the quality of the MatNet solution is far superior. With instance augmentation ($\times 128$), MatNet's optimality gap goes down to 0.01% for 20-city ATSP, or to less than 1% for 100-city ATSP.

## 5 Flexible flow shop problem

**Problem definition.** Flexible flow shop problem (FFSP), or sometimes called hybrid flow shop problem (HFSP), is a classical optimization problem that captures the complexity of the production scheduling processes in real manufacturing applications. It is defined with a set of $N$ jobs that has to be processed in $S > 1$ stages all in the same order. Each stage consists of $M > 1$ machines,[5] and a job can be handled by any of the machines in the same stage. A machine cannot process more than one job at the same time. The goal is to schedule the jobs so that all jobs are finished in the shortest time possible (*i.e.*, minimum makespan).

For our experiment, we fix on a configuration of stages and machines of a moderate complexity. We assume that there are $S = 3$ stages, and each stage has $M = 4$ machines. At the $k$th stage, the processing time of the job $j$ on the machine $i$ is given by $\mathbf{D}_{ij}^{(k)}$. Therefore, an instance of the problem is defined by three processing time matrices ($\mathbf{D}^{(1)}, \mathbf{D}^{(2)}$, and $\mathbf{D}^{(3)}$), all of which have the size $M$-by-$N$. $\mathbf{D}_{ij}^{(k)}$ is filled with a random integer between 2 and 9 for all $i$, $j$, and $k$. We solve three

---

[5]In general, the number of machines in each stage can vary from stage to stage. When all stages have just one machine, the problem simplifies to flow shop problem.

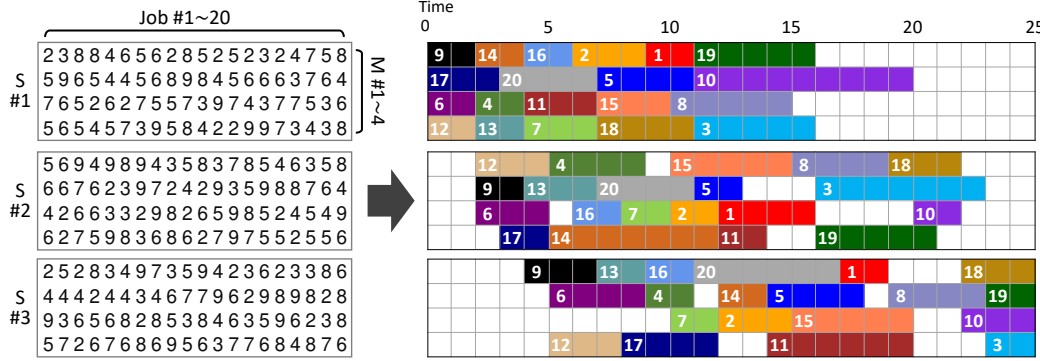

Figure 4: (LEFT) Processing time matrices for a 3-stage 4-4-4-machine 20-job flexible flow shop problem. (RIGHT) A Gantt chart showing a valid schedule for the given problem. Numbers on colored strips are assigned job indices. This schedule has been produced by our MatNet-based neural net model, and it has a makespan of 25.

types of FFSP with different number of jobs, namely, $N$ =20, 50, and 100. An example instance of FFSP with job count $N = 20$ is shown in Figure 4 along with a Gantt chart showing an example of a valid schedule.

**Encoding & decoding.** To encode three processing time matrices, $\mathbf{D}^{(k)}(k = 1, 2, 3)$, we use 3 copies of a MatNet model, one for each matrix, and acquire vector representations $\hat{h}_{a_i}^{(k)}|_{i=1,2,3,4}$ for machines and $\hat{h}_{b_j}^{(k)}|_{j=1,2,\cdots,20}$ for jobs in stage $k$. This stage-wise encoding scheme is the key to develop a proper scheduling strategy, which should follow different job-selection rules depending on whether you are in an early stage or a late one. All three MatNets are configured identically, using the same hyperparameters as the ATSP encoder explained in the previous section.[6] Machines are initially embedded with one-hot vectors ($M_{\max} = 4$), and jobs are embedded with zero vectors. We also use three decoding neural nets, $\mathcal{G}_k(k = 1, 2, 3)$, all of them having the same architecture as the ATSP decoder (Figure 3).

Generating an FFSP solution means completing a Gantt chart like the one in Figure 4. We start at the upper left corner of the chart (time $t = 0$) and move from stage 1 to stage 3. At stage $k$, we loop over its 4 machines $i = 1, 2, 3, 4$, each time using $\hat{h}_{a_i}^{(k)}$ as the QUERY token for $\mathcal{G}_k$. The input embeddings for $\mathcal{G}_k$ are $\hat{h}_{b_j}^{(k)}$ ($j = 1, 2, \cdots, 20$), plus one extra (21st) job embedding made of learnable parameters for "skip" option. $\mathcal{G}_k$ outputs selection probabilities for jobs $j = 1, 2, \cdots, 20$, and 21. When the "skip" option is selected, no job is assigned to machine $i$ even if there are available jobs. This is sometimes necessary to create better schedules. Obviously, unavailable jobs (already processed or not ready for stage $k$) are masked in the decoder. When no job can be selected (no job at stage $k$, or the machine $i$ is busy), we simply do not assign any job to the machine and move on. When we finish assigning jobs for all machines, we increment time $t$ and repeat until all jobs are finished. See Appendix J for the flow of the FFSP decoding process.

**Training** The decoding process is a series of job assignments to machines in each stage, but note that the order of the machines we assign jobs to can be arbitrary. We use $4! = 24$ number of permutations of (#1, #2, #3, #4) for the order of machines in which we execute job assignments at each stage. From the same set of vector representations of machines ($\hat{h}_{a_i}^{(k)}$) and jobs ($\hat{h}_{b_j}^{(k)}$), this permutation trick can create 24 heterogeneous trajectories (different schedules), which we use for the POMO reinforcement learning to train the networks. We use a batch size 50 and Adam optimizer with a learning rate of $1 \times 10^{-4}$. One epoch being processing 1,000 instances, we train for (100, 150, 200) epochs for FFSP with job size $N = (20, 50, 100)$, which takes (4, 8, 14) hours. For FFSP with $N = 100$, we accumulate gradients from 4 GPUs.

---

[6]except for the number of encoding layers $L = 3$, descreased from 5. This change is optional, but we have found that $L = 5$ is unnecessarily too large for the task.

Table 5.2: Experiment results on 1,000 instances of FFSP

| Method | FFSP20 | | | FFSP50 | | | FFSP100 | | |
|---|---|---|---|---|---|---|---|---|---|
| | MS | Gap | Time | MS | Gap | Time | MS | Gap | Time |
| CPLEX (60s) | 46.4 | 21.0 | (17h) | × | | | × | | |
| CPLEX (600s) | 36.6 | 11.2 | (167h) | | | | | | |
| Random | 47.8 | 22.4 | (1m) | 93.2 | 43.6 | (2m) | 167.2 | 77.5 | (3m) |
| Shortest Job First | 31.3 | 5.9 | (40s) | 57.0 | 7.4 | (1m) | 99.3 | 9.6 | (2m) |
| Genetic Algorithm | 30.6 | 5.2 | (7h) | 56.4 | 6.8 | (16h) | 98.7 | 9.0 | (29h) |
| Particle Swarm Opt. | 29.1 | 3.7 | (13h) | 55.1 | 5.5 | (26h) | 97.3 | 7.6 | (48h) |
| MatNet | 27.3 | 1.9 | (8s) | 51.5 | 1.9 | (14s) | 91.5 | 1.8 | (27s) |
| MatNet ($\times$128) | 25.4 | - | (3m) | 49.6 | - | (8m) | 89.7 | - | (23m) |

**Inference.** We use sampled solutions and the POMO inference algorithm with 24 machine-order permutations similarly to the training. Interestingly, the sampled solutions are significantly better than the greedily-chosen ones, which could be a consequence of the stochastic nature of the problem in our approach. Each decoder $\mathcal{G}_k$ makes decisions without any knowledge of other ($\neq k$) stages.

## 5.1 Comparison with other baselines

In Table 5.2, we record average makespan (MS) of the schedules produced by various optimization methods for the same 1,000 test instances. The gaps are given in absolute terms with respect to MatNet with $\times$128 augmentation result. We display the runtime of each method, following the same rules that we use for Table 4.1 (*e.g.*, computation time only, scaling by $1/8$ for single-thread processes, and single-GPU inference for the MatNet-based method).

**Mixed-integer programming (MIP).** Modeling flow shop problems using MIP is possible, but it is much more complex than for TSPs. We use an FFSP model found in literature [33] with modifications that (empirically) improves the search speed of CPLEX. Even with the modifications, however, CPLEX cannot produce optimal solutions for any of the test instances within a reasonable time. For FFSPs with job size $N = 20$, we show two results, recorded with timeouts of 60 and 600 seconds per instance. For $N = 50$ and 100, no valid solutions are found within 600 seconds.

**(Meta-)Heuristics.** Random and Shortest-Job-First (SJF) methods are greedy heuristics. They create valid schedules in an one-shot manner using the Gantt-chart-completion strategy, similarly to our MatNet based model. SJF assigns the shortest jobs (at most 4) in ascending order that are available for each stage at each time step $t$. The runtimes for Random and SJF are slower than that of MatNet in Table 5.2 because MatNet solves a batch of problems in parallel using GPUs.

Genetic Algorithm (GA) and Particle Swarm Optimization (PSO) are two metaheuristics widely used by the OR community to tackle the FFSP. Metaheuristics are systematic procedures to create heuristics, most useful for optimization problems that are too complex to use MIP approaches or to engineer problem-specific handcrafted algorithms. Our implementation of GA and PSO are based on the works found in the OR literature [34, 35].

**MatNet.** Solutions produced by the learned heuristic of our MatNet based model significantly outperform those of the conventional OR approaches, both in terms of the qualities and the runtimes. Some commercial applications of the FFSP require solving just one instance as best as one can within a relatively generous time budget. Hence, we have also tested the performance of baseline algorithms under ample time to improve their solutions on single instances. The results are provided in Appendix I. Even in this case, however, the conventional OR algorithms do not produce better solutions than those of our MatNet method.

# 6    Conclusion and Discussion

In this paper, we have introduced MatNet, a neural net capable of encoding matrix-style relationship data found in many CO problems. Using MatNet as a front-end model, we have solved two classical optimization problems of different nature, the ATSP and the FFSP, for the first time using a deep learning approach.

A neural heuristic solver that clearly outperforms conventional OR methods has been rare, especially for classical optimization problems. Perhaps, this is because the range of CO problems ML researchers attempt to solve has been too narrow, only around simple problems for which good heuristics and powerful MIP models already exist (such as the TSP and other related routing problems). The FFSP is not one of them, and we have shown that our MatNet-based FFSP solver significantly outperforms other algorithms. Our results are promising, hinting the prospect of real-world deployments of neural CO solvers in the near future. More research is needed, however, to fully reflect combinations of many different types of constraints posed by real-world problems.

For our experiments, we have chosen to use an end-to-end RL approach for its simplicity. It is purely data-driven and does not require any engineering efforts by a domain expert. We would like to emphasize, however, that the use of MatNet is not restricted to end-to-end methods only. Hybrid models (ML + OR algorithms) based on the MatNet encoder should be possible, and they will have better performance and broader applicability.

We have implemented our MatNet model in PyTorch. The training and testing code for the experiments described in the paper is publicly available.[7]

---

[7] https://github.com/yd-kwon/MatNet

## Acknowledgments and Disclosure of Funding

We thank anonymous reviewers for their valuable comments that helped improve the paper. We thank Kevin Tierney for reviewing the MIP methods used in our experiments and giving us helpful tips and comments. We declare no third party funding or support.

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
