# A MatNet variants

## A.1 Multiple data matrices

A combinatorial optimization problem can be presented with multiple ($f$) relationship features between two groups of items. In FFSP, for example, a production cost could be different for each process that one has to take into account for scheduling in addition to the processing time for each pair of the job and the machine. (The optimization goal in this case would be to minimize the weighted sum of the makespan and the total production cost.) When there are $f$ number of matrices that need to be encoded ($\mathbf{D}^1, \mathbf{D}^2, \ldots, \mathbf{D}^f$), MatNet can be easily expended to accommodate such problems by using the mixed-score attention shown in Figure A.1 instead of the one in Figure 2(b). "Trainable element-wise function" block in Figure A.1 is now an MLP with $f + 1$ input nodes and 1 output node.

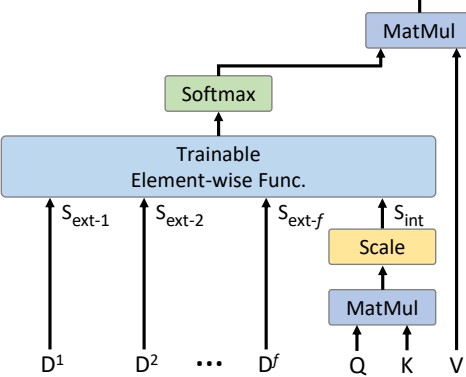

Figure A.1: Mixed-score attention with $f$ number of data matrices ($\mathbf{D}^1, \mathbf{D}^2, \ldots, \mathbf{D}^f$).

## A.2 Alternative encoding sequences

Equation (2) in the main text describes the application of $\mathcal{F}_A$ and $\mathcal{F}_B$ in the graph attentional layer of MatNet that happens in parallel. One can change it to be sequential, meaning that $\mathcal{F}_A$ is applied first and then the application of $\mathcal{F}_B$ follows using the updated vector representations $\hat{h}'_{a_i}$. This is illustrated in Figure A.2. The opposite ordering is also valid, in which $\mathcal{F}_B$ is applied first and then $\mathcal{F}_A$ follows (not drawn).

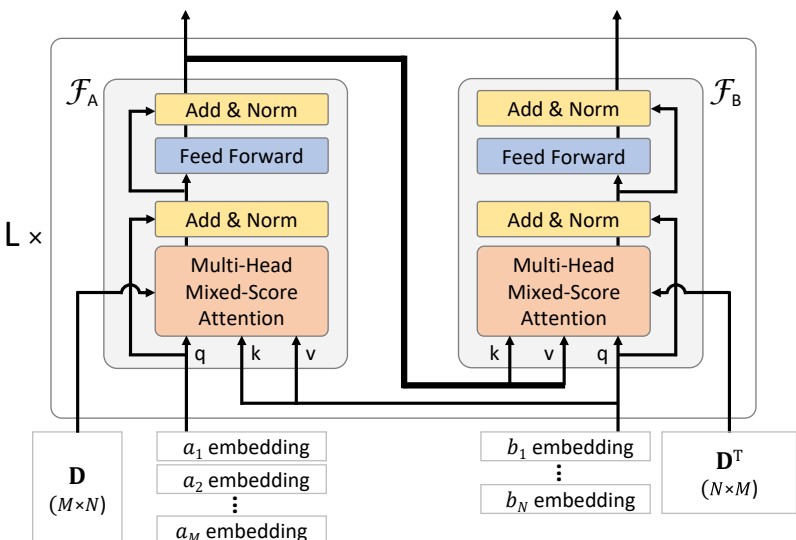

Figure A.2: An alternative MatNet structure with the sequential update scheme. The bold line indicates the change from the original.

It is not yet clear why, but we have empirically found that the alternative MatNet structure above leads to better-performing models than the original structure in some experiments (that are not the ATSP and the FFSP experiments described in the paper).

## A.3 Alternatives to one-hot initial node embeddings

For initial node representations of nodes in group B, one only needs mutually distinct-enough $N_{\max}$ vectors and they do not need to be of one-hot type. We have chosen to present our model with one-hot vectors because it is the simplest to implement this way. But for the readers who look for more generalizability, this may seem too restrictive.

Instead of one-hot vectors, one can use $N_{\max}$ different vectors made of learnable parameters (just like the parameters of the neural net). They automatically become mutually-distinct (well-performing) vectors during the model training. This way, one can use arbitrarily large $N_{\max}$, without being limited by the pre-defined length of the embedding vectors.

Ultimately, one can use vectors that are created randomly for each problem instance, which completely lift the restriction of $N_{\max}$. Although this leads to slightly worse performance compared to the models with one-hot initial node embeddings (See Figure A.3), initial node embedding with random vectors can be immensely useful in situations when settling on a fixed value for $N_{\max}$ is undesirable.

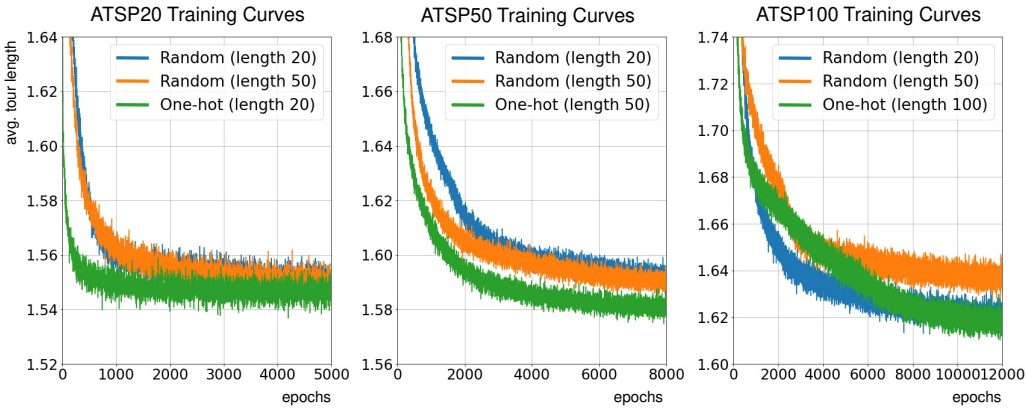

Figure A.3: Random vector (length 20 and 50) vs. one-hot vector initial node embeddings

## A.4 Single update function, $\mathcal{F} = \mathcal{F}_A = \mathcal{F}_B$

Rather than having two separate update functions $\mathcal{F}_A$ and $\mathcal{F}_B$ (Section 3.1), MatNet can be implemented to use a single update function for all nodes, both in $A$ and $B$, treating them all equally. (In other words, $\mathcal{F}_A$ and $\mathcal{F}_B$ will share the same parameter set.) This alternate approach has an advantage that the model size of MatNet is reduced to almost a half.

Table A.1 compares the original MatNet result on ATSP and FFSP (those presented in the main text) and the results using the modified MatNet with a single update function. Experiments using the modified MatNet have been performed under the same training and inference conditions, including the number of training epochs. It is shown that, especially for CO problems for which item groups $A$ and $B$ are of the similar type (*e.g.*, ATSP), this change in the model architecture has a relatively small negative effect on the solver's performance.

We note, however, that the use of single update function does not reduce the training and inference speed or the GPU memory usage, the computing resources that are usually considered more valuable than the (saved) model size. Also, presenting MatNet with two separate update functions as we have done so in the main text allows it to be modified and expanded more easily for other related CO problems with two or more groups of items.

Table A.1: Experiment results using dual (original) vs. single update function

|  | ATSP50 | ATSP100 | FFSP50 | FFSP100 |
|---|---|---|---|---|
| Original MatNet | 1.580 | 1.621 | 51.545 | 91.529 |
| Original MatNet ($\times$128) | 1.561 | 1.585 | 49.625 | 89.701 |
| Single-$\mathcal{F}$ MatNet | 1.580 | 1.623 | 51.574 | 91.680 |
| Single-$\mathcal{F}$ MatNet ($\times$128) | 1.561 | 1.587 | 49.649 | 89.884 |

# B ATSP definition and baselines

## B.1 Tmat class

While one can generate an ATSP instance by simply filling a distance matrix with random numbers alone (an "amat" class problem), such problem lacks the correlation between distances and is uninteresting. In our ATSP experiments, we use "tmat" class ATSP instances [27] that have the triangle inequality. First, we populate the distance matrix with independent random integers between 1 and $10^6$, except for the diagonal elements $d(c_i, c_i)$ that are set to 0. If we find $d(c_i, c_j) > d(c_i, c_k) + d(c_k, c_j)$ for any $(i, j, k)$, we replace it with $d(c_i, c_j) = d(c_i, c_k) + d(c_k, c_j)$. This procedure is repeated until no more changes can be made.

Generation speed of tmat class instances can be greatly enhanced using parallel processing on GPUs. Below is the Python code for the instance generation using Pytorch library.

```python
def generate(batch_size, problem_size, min_val=1, max_val=1000000):

    problems = torch.randint(low=min_val, high=max_val+1, size=(batch_size,
        problem_size, problem_size))
    problems[:, torch.arange(problem_size), torch.arange(problem_size)] = 0

    while True:
        old_problems = problems.clone()

        problems, _ = (problems[:, :, None, :] + problems[:, None, :,
            :].transpose(2,3)).min(dim=3)

        if (problems == old_problems).all():
            break

    return problems
```

## B.2 MIP model

Our MIP model for ATSP is based on Miller-Tucker-Zemlin [30] developed in 1960.

**Indices**
$i, j$      City index

**Parameters**
$n$      Number of Cities
$c_{ij}$      Distance from city $i$ to city $j$

**Decision variables**
$x_{ij}$      $\begin{cases} 1 & \text{if you move from city } i \text{ to city } j \\ 0 & \text{otherwise} \end{cases}$

$u_i$      arbitrary numbers representing the order of city $i$ in the tour

**Objective:**

$$\text{minimize}\left( \sum_{i=1}^{n} \sum_{j=1}^{n} c_{ij} x_{ij} \right) \tag{B.3}$$

**Subject to:**

$$\sum_{i=1}^{n} x_{ij} = 1 \qquad j = 1, 2, \cdots, n \tag{B.4}$$

$$\sum_{j=1}^{n} x_{ij} = 1 \qquad i = 1, 2, \cdots, n \tag{B.5}$$

$$u_i - u_j + (n-1) \cdot x_{ij} \le n - 2 \qquad i, j = 2, \cdots, n \tag{B.6}$$

Constraints (B.4) and (B.5) enforce the one-visit-per-city rule. Constraint set (B.6) could look unintuitive, but it is used to prevent subtours so that all cities are contained in a single tour of length $n$ (commonly known in the OR community as MTZ subtour elimination constraints).

### B.3 Heuristics

**Greedy-selection heuristics** Implementation of Nearest Neighbor (NN) is self-explanatory. On the other hand, exact procedures for the insertion-type heuristics (NI and FI) can vary when applied to the ATSP. Our approach is the following: for each city $c$ that is not included in the partially-completed round-trip tour of $k$ cities, we first determine the insertion point (one of $k$ choices) that would make the insertion of the city $c$ into the tour increase the tour length the smallest. With the insertion point designated for each city, every city now has the increment value for the tour length associated with it. Nearest Insertion (NI) selects the city with the smallest increment value, and Furthest Insertion (FI) selects the one with the largest increment value.

**LKH3** The version of LKH3 that we use is 3.0.6. The source code is downloaded from `http://webhotel4.ruc.dk/~keld/research/LKH-3/LKH-3.0.6.tgz`. Parameter `MAX_TRIALS` is set to $2 \times N$, where $N$ is the number of cities. Parameter `RUNS` is set to 1. All other parameters are set to the default values.

## C   Instance augmentation vs. sampling

The instance augmentation technique effectively creates different problem instances from which the neural net generates solutions of great diversity [8]. In Table C.1, we compare the performance of the instance augmentation method with that of the sampling method on the ATSP of different sizes. (The first two data rows of Table C.1 are identical to the last two rows of Table 1 in the main text.) The instance augmentation method takes a bit more time than the sampling method when the two use the same number (128) of rollouts because the former requires a new encoding procedure for each rollout while the latter needs to run the encoder just once and then reuses its output repeatedly. The table shows that the sampling method clearly suffers from the limited range of solutions it can create. Even when the number of rollouts used is larger by a factor of 10, the sampling method cannot outperform the instance augmentation technique in terms of the solution quality.

Table C.1: Experiment results on 10,000 instances of ATSP using different inference methods.

| Method | ATSP20 | | | ATSP50 | | | ATSP100 | | |
|---|---|---|---|---|---|---|---|---|---|
| | Len. | Gap | Time | Len. | Gap | Time | Len. | Gap | Time |
| MatNet (single POMO) | 1.55 | 0.53% | (2s) | 1.58 | 1.34% | (8s) | 1.62 | 3.24% | (34s) |
| MatNet ($\times$128 inst. aug.) | **1.54** | **0.01**% | (4m) | **1.56** | **0.11**% | (17m) | **1.59** | **0.93**% | (1h) |
| MatNet ($\times$128 sampling) | 1.54 | 0.22% | (1m) | 1.57 | 0.52% | (7m) | 1.60 | 1.89% | (37m) |
| MatNet ($\times$1280 sampling) | 1.54 | 0.15% | (12m) | 1.57 | 0.37% | (1h) | 1.60 | 1.58% | (6h) |

# D Euclidean TSP Experiment

Many previous neural approaches on solving TSP assume Euclidean distance and use x, y coordinates of the cities as the input. To compare performance of our MatNet-based (general) TSP solver with those of the others, we test our model on (symmetric) distance matrices that are created from the same list of x, y coordinates on Euclidean space used by the other methods. Strictly speaking, other neural net based TSP solvers have the unfair advantage in this comparison, because they have the hard-coded prior on the distribution of the the problem instances (*i.e.*, they are all assumed to have a distance measure that applies uniformly to all pairs of cities). We use exactly the same model used for the asymmetric TSP experiments presented in the main text, only training it with different data (symmetric distance matrices) this time. Its performance on TSP50 and TSP100 are presented in Table D.1.

One of the important baselines included in the table is the AM model trained by POMO algorithm [8]. Similarly to our MatNet-based approach, it is also a construction-type method and has Transformer-based architecture. (But unlike MatNet, the instance augmentation for the AM is limited to ×8 and cannot be made larger.) Despite the fact that MatNet is structurally designed to handle much broader range of TSP problems while the AM specializes in solving only the Euclidean ones, MatNet-based Euclidean TSP solver shows performance that is still competitive, demonstrating its wide adaptability.

Table D.1: Experiment results on 10,000 instances of Euclidean TSP

| Method | TSP50 | | | TSP100 | | |
|---|---|---|---|---|---|---|
| | Len. | Gap | Time | Len. | Gap | Time |
| Concorde [36] (Optimal) | 5.69 | - | (2m) | 7.76 | - | (8m) |
| GCN-BS [37] | 5.69 | 0.01% | (18m) | 7.87 | 1.39% | (40m) |
| 2-Opt-DL [38], 2K | 5.70 | 0.12% | (29m) | 7.83 | 0.87% | (41m) |
| LIH [39], 5K | 5.70 | 0.20% | (1h) | 7.87 | 1.42% | (2h) |
| AM + POMO | 5.70 | 0.21% | (2s) | 7.80 | 0.46% | (11s) |
| AM + POMO (×8) | 5.69 | 0.03% | (16s) | 7.77 | 0.14% | (1m) |
| MatNet | 5.71 | 0.30% | (8s) | 7.83 | 0.94% | (34s) |
| MatNet (×8) | 5.69 | 0.05% | (1m) | 7.79 | 0.41% | (5m) |
| MatNet (×128) | 5.69 | 0.01% | (16m) | 7.78 | 0.17% | (1h) |

# E Note on the graph embedding in the `QUERY` token

Ablation studies show that including "the graph embedding" (the average of the embedding vectors of all the nodes in the graph) as a part of the `QUERY` token has a negligible effect in our ATSP solver. The same is observed when tested with the AM-based TSP solvers [8] as well.

Note that dropping "the graph embedding" from the `QUERY` token does not necessarily mean that the view of the solver is now restricted to a local one, or any less than that of the solver equipped with a QUERY containing "the graph embedding." The attention mechanism makes a weighted sum of the "values." If all the weights are made the same (via constant values used in creating "keys" and "queries") and if the "values" become just the copies of the representations of the nodes (via identity transformation), the output of the attentional layer is the sum of the representations of all the nodes, which is "the graph embedding." Therefore, it is possible to contain "the graph embedding" information in the embedding of any node, if the reinforcement learning process finds it necessary.

# F Training curves

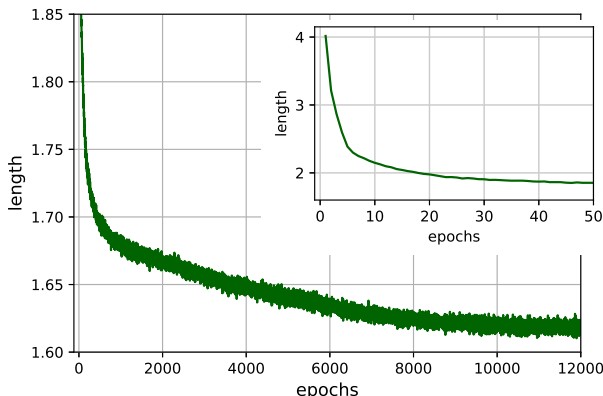

Figure F.1: The training curve of the MatNet-based ATSP solver with 100-city instances.

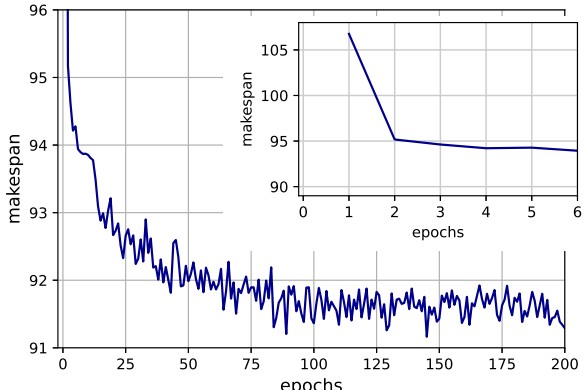

Figure F.2: The training curve of the MatNet-based FFSP solver with 100-job instances.

# G FFSP definition and baselines

## G.1 Unrelated parallel machines

We choose to generate processing time matrices with random numbers. This makes machines in the same stage totally unrelated to each other. This is somewhat unrealistic because an easy (short) job for one machine tends to be easy for the others, and the sames goes for a difficult (long) job, too. The type of FFSP that is studied most in the OR literature is a simpler version, in which all machines are identical ("uniform parallel machines" [40]). Solving "unrelated parallel machines" FFSP is more difficult, but the algorithms designed for this type of FFSP can be adapted to real-world problems more naturally than those assuming identical machines.

## G.2 MIP model

Our MIP model for FFSP is (loosely) based on Asadi-Gangraj [33] but improved to perform better when directly entered into the CPLEX platform.

**Indices**

$i$        Stage index

$j, l$       Job index

$k$       Machine index in each stage

$n$       Number of jobs

$m$       Number of stages

**Parameters**

$S_i$       Number of machines in stage $i$

$M$       A very large number

$p_{ijk}$     Processing time of job $j$ in stage $i$ on machine $k$

**Decision variables**

$C_{ij}$       Completion time of job $j$ in stage $i$

$X_{ijk}$     $\begin{cases} 1 & \text{if job } j \text{ is assigned to machine } k \text{ in stage } i \\ 0 & \text{otherwise} \end{cases}$

$Y_{ilj}$     $\begin{cases} 1 & \text{if job } l \text{ is processed earlier than job } j \text{ in stage } i \\ 0 & \text{otherwise} \end{cases}$

**Objective:**

$$\text{minimize} \left( \max_{j=1..n} \{C_{mj}\} \right) \tag{G.7}$$

**Subject to:**

$$\sum_{k=1}^{S_i} X_{ijk} = 1 \quad \begin{cases} i = 1, 2, \cdots, m \\ j = 1, 2, \cdots, n \end{cases} \tag{G.8}$$

$$Y_{ijl} = 0 \quad \begin{cases} i = 1, 2, \cdots, m \\ j = 1, 2, \cdots, n \end{cases} \tag{G.9}$$

$$\sum_{j=1}^{n} \sum_{l=1}^{n} Y_{ijl} = \sum_{k=1}^{S_i} \max \left( \sum_{j=1}^{n} (X_{ijk}) - 1, \ 0 \right) \quad i = 1, 2, \cdots, m \tag{G.10}$$

$$Y_{ijl} \leq \max \left( \max_{k=1..S_i} \{X_{ijk} + X_{ilk}\} - 1, \ 0 \right) \quad \begin{cases} i = 1, 2, \cdots, m \\ j, l = 1, 2, \cdots, n \end{cases} \tag{G.11}$$

$$\sum_{l=1}^{n} Y_{ijl} \leq 1 \quad \begin{cases} i = 1, 2, \cdots, m \\ j = 1, 2, \cdots, n \end{cases} \tag{G.12}$$

$$\sum_{l=1}^{n} Y_{ilj} \leq 1 \quad \begin{cases} i = 1, 2, \cdots, m \\ j = 1, 2, \cdots, n \end{cases} \tag{G.13}$$

$$C_{1j} \geq \sum_{k=1}^{S_1} p_{1jk} \cdot X_{1jk} \quad j = 1, 2, \cdots, n \tag{G.14}$$

$$C_{ij} \geq C_{i-1j} + \sum_{k=1}^{S_i} p_{ijk} \cdot X_{ijk} \quad \begin{cases} i = 2, 3, \cdots, m \\ j = 1, 2, \cdots, n \end{cases} \tag{G.15}$$

$$C_{ij} + M(1 - Y_{ilj}) \geq C_{il} + \sum_{k=1}^{S_i} p_{ijk} \cdot X_{ijk} \quad \begin{cases} i = 1, 2, \cdots, m \\ j, l = 1, 2, \cdots, n \end{cases} \tag{G.16}$$

Constraint set (G.8) ensures that each job must be assigned to one machine at each stage. Constraint sets (G.9)–(G.13) define precedence relationship ($Y$) between jobs within a stage. Constraint set (G.9) indicates that every job has no precedence relationship ($Y = 0$) with itself. Constraint set

(G.10) indicates that the sum of all precedence relationships (the sum of all $Y$) in a stage is the same as $n - S_i$ minus the number of machines that no job has been assigned. Constraint set (G.11) expresses that only the jobs assigned to the same machine can have precedence relationships ($Y = 1$) among themselves. Constraint sets (G.12) and (G.13) mean that a job can have at most one preceding job and one following job. Constraint set (G.14) indicates that completion time of job $j$ in the first stage is greater than or equal to its processing time in this stage. The relation between completion times in two consecutive stages for job $j$ can be seen in Constraint set (G.15). Constraint set (G.16) guarantees that no more than one job can run on the same machine at the same time.

### G.3 Metaheuristics

**Genetic algorithm**  Genetic algorithm (GA) iteratively updates multiple candidate solutions called chromosomes. Child chromosomes are generated from two parents using crossover methods, and mutations are applied on chromosomes for better exploration.

Our implementation of GA is based on chromosomes made of $S \times N$ number of real numbers, where $S$ is the number of stages, and $N$ is the number of jobs. Each real number within a chromosome corresponds to the scheduling of one job at one stage. The integer part of the number determines the index of the assigned machine, and the fractional part determines the priority among the jobs when there are multiple jobs simultaneously available. The integer and the fractional parts are independently inherited during crossover. For mutation, we randomly select one from the following four methods: exchange, inverse, insert, and change. The number of chromosomes we use is 25, and the crossover ratio and the mutation rate are both 0.3. One of the initial chromosomes is set to the solution of the SJF heuristic and the best-performing chromosome is conserved throughout each iteration. We run 1,000 iterations per instance.

**Particle swarm optimization**  Particle swarm optimization (PSO) is a metaheuristic algorithm that iteratively updates multiple candidate solutions called particles. Particles are updated by the weighted sum of the inertial value, the local best and the global best at each iteration.

Our PSO solution representation (a particle) has the same form as a chromosome of our GA implementation. We use 25 number of particles for PSO. The inertial weight is 0.7, and the cognitive and social constants are set to 1.5. One of the initial particles is made to represent the solution of the SJF heuristic. We run 1,000 iterations per instance.

## H  Generalization performance on FFSP

A trained MatNet model can encode a matrix of an arbitrary number of rows (or columns). Such characteristics of MatNet is most useful for problems like FFSP, in which one of the two groups of items that the problem deals with is frequently updated. If we imagine a factory that is in need for an optimization tool for the FFSP-type scheduling problems, it is likely that its schedules are updated every day (or even every hour) or so, each time with a different (various sizes) set of jobs. The set of machines in the schedule, on the other hand, is unlikely to change on a daily basis. The processing time matrices in this case have rows of an unspecified size but a fixed number of columns. A MatNet-based FFSP solver can naturally handle such data.

In Table H.1, we show the performance of the three MatNet-based FFSP solvers. They are the same models that are used for the FFSP experiments described in the main text, *e.g.*, in Table 2. Each model is trained with the FFSP instances of a different number of jobs ($N_{\text{train}} = 20, 50, 100$). We test them with 1,000 instances of the FFSP of different job sizes ($N_{\text{test}} = 20, 50, 100, 1,000$), and the average makespans are shown in the table. We have used $\times 128$ instance augmentation for inference. Notice that the MatNet-based models can handle $N_{\text{test}} = 1,000$ cases reasonably well, even though they have not encountered such large instances during the training. SFJ results are displayed as a baseline.

Table H.1: Generalization test results on 1,000 instances of FFSP.

| Method | FFSP20 MS | FFSP50 MS | FFSP100 MS | FFSP1000 MS |
|---|---|---|---|---|
| Shortest Job First | 31.3 | 57.0 | 99.3 | 847.0 |
| MatNet ($N_{\text{train}} = 20$) | 25.4 | 50.3 | 91.2 | 814.4 |
| MatNet ($N_{\text{train}} = 50$) | **25.2** | 49.6 | 89.9 | 803.9 |
| MatNet ($N_{\text{train}} = 100$) | 25.3 | **49.6** | **89.7** | **803.2** |

# I  One-instance FFSP

A scheduling task in a factory does not require solving many different problem instances at once. Moreover, the runtime is usually allowed quite long for the scheduling. We test each baseline algorithm and our MatNet-based model on a single FFSP instance and see how much each algorithm can improve its solution quality within a reasonably long time. For each FFSP size ($N = 20, 50, 100$), one problem instance is selected manually from 1,000 saved test instances based on the results from the "fast scheduling" experiments (in Table 2 of the main text). We have chosen instances whose makespans roughly match the average makespans of all test instances. The processing time matrices for the selected $N = 20, 50$ and 100 instances are displayed in Figure 4, Figure I.1, and Figure I.2, respectively.

Table I.1 shows the result of the one-instance experiments.[8] For both the MIP and the metaheuristic approaches, we find that only a relatively small improvement is possible even when they are allowed to keep searching for many more hours. MatNet approach produces much better solutions in the order of seconds.[9]

Table I.1: One-instance FFSP experiment results.

| Method | FFSP20 (Fig.4) MS | Time | FFSP50 (Fig.G.1) MS | Time | FFSP100 (Fig.G.2) MS | Time |
|---|---|---|---|---|---|---|
| CPLEX (10 min timeout) | 37 | (10m) | $\times$ | | $\times$ | |
| CPLEX (10 hour timeout) | 34 | (10h) | | | | |
| Shortest Job First | 31 | (-) | 58 | (-) | 100 | (-) |
| GA ($10^3$ iters, 1 run) | 30 | (4m) | 57 | (8m) | 99 | (14m) |
| GA ($10^5$ iters, 1 run) | 30 | (8h) | 57 | (13h) | 99 | (26h) |
| GA ($10^3$ iters, 128 run) | 29 | (1h) | 57 | (2h) | 99 | (4h) |
| PSO ($10^3$ iters, 1 run) | 29 | (7m) | 56 | (14m) | 98 | (24m) |
| PSO ($10^5$ iters, 1 run) | 27 | (12h) | 56 | (21h) | 98 | (40h) |
| PSO ($10^3$ iters, 128 run) | 27 | (2h) | 55 | (4h) | 98 | (7h) |
| MatNet (single POMO) | 27 | (2s) | 51 | (4s) | 93 | (6s) |
| MatNet ($\times 128$ inst. aug.) | 25 | (5s) | 50 | (10s) | 91 | (11s) |
| MatNet ($\times 1280$ inst. aug.) | 25 | (8s) | 49 | (16s) | 90 | (23s) |

---

[9]The runtimes for GA and PSO are divided by 8, following the convention used in the main text, only for 128 run cases. Single-instance single-run programs cannot easily utilize multiprocessing.

[9]Here, for MatNet approaches, we simply use the instance augmentation technique only. There are, however, better inference techniques for neural net based approaches that are more suitable for solving "one-instance CO problems." See, for example, Hottung *et al.* [41].

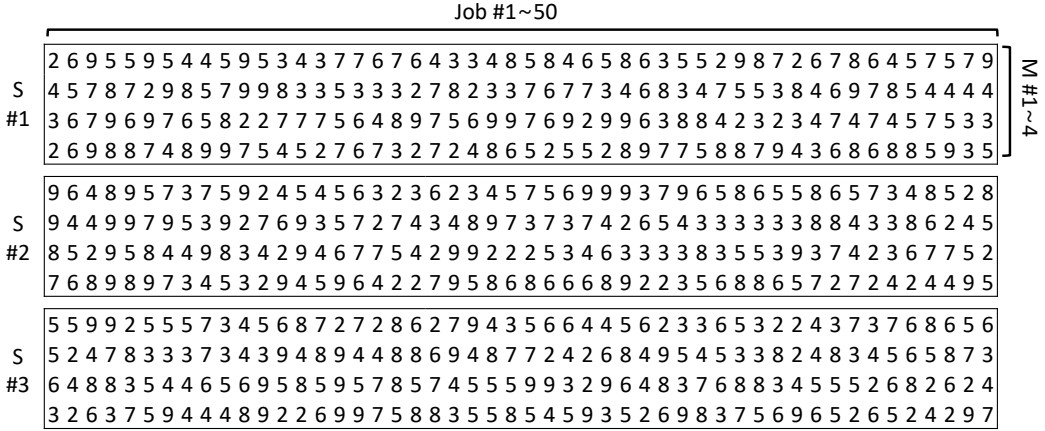

Figure I.1: FFSP50 processing time matrices used for the one-instance experiment.

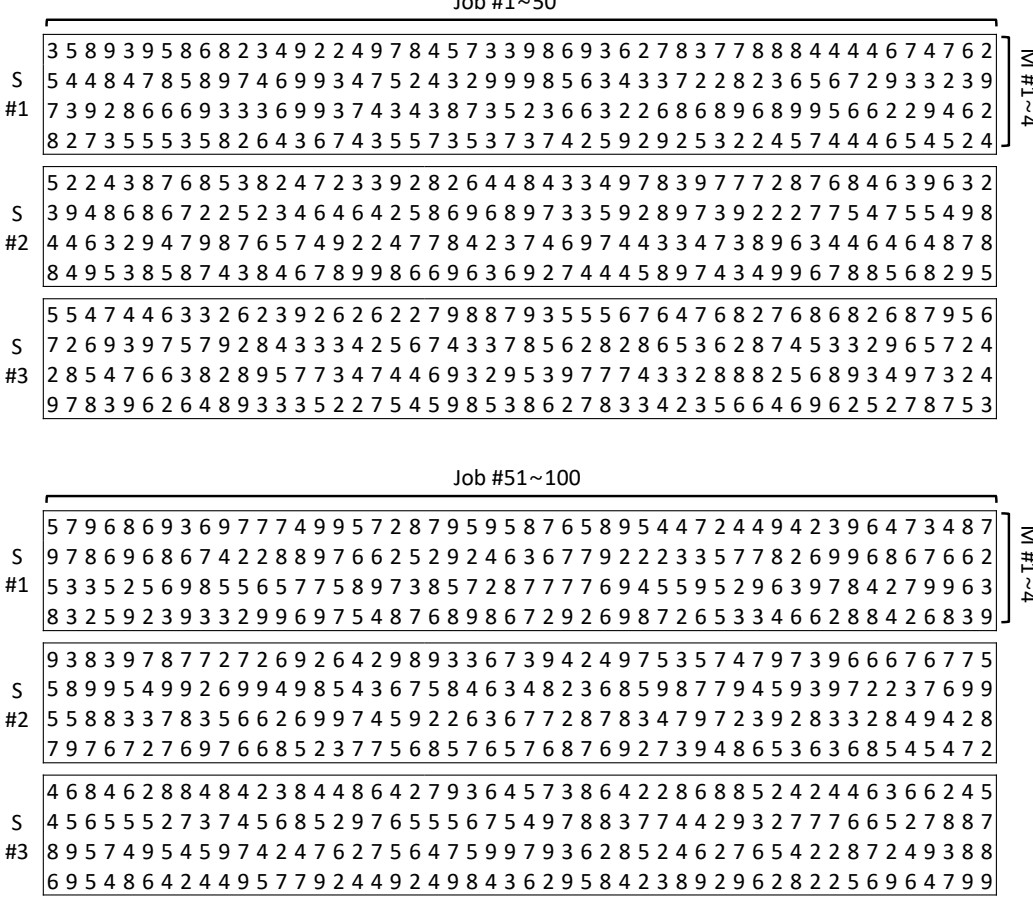

Figure I.2: FFSP100 processing time matrices used for the one-instance experiment.

## J    Diagrams for ATSP and FFSP solvers

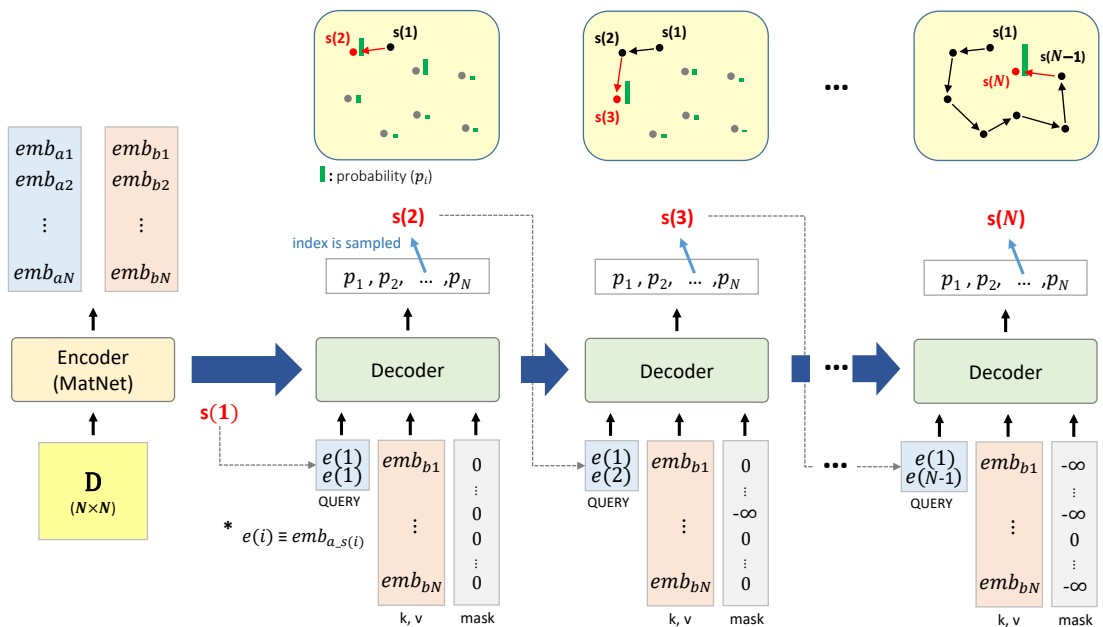

Figure J.1: ATSP solver

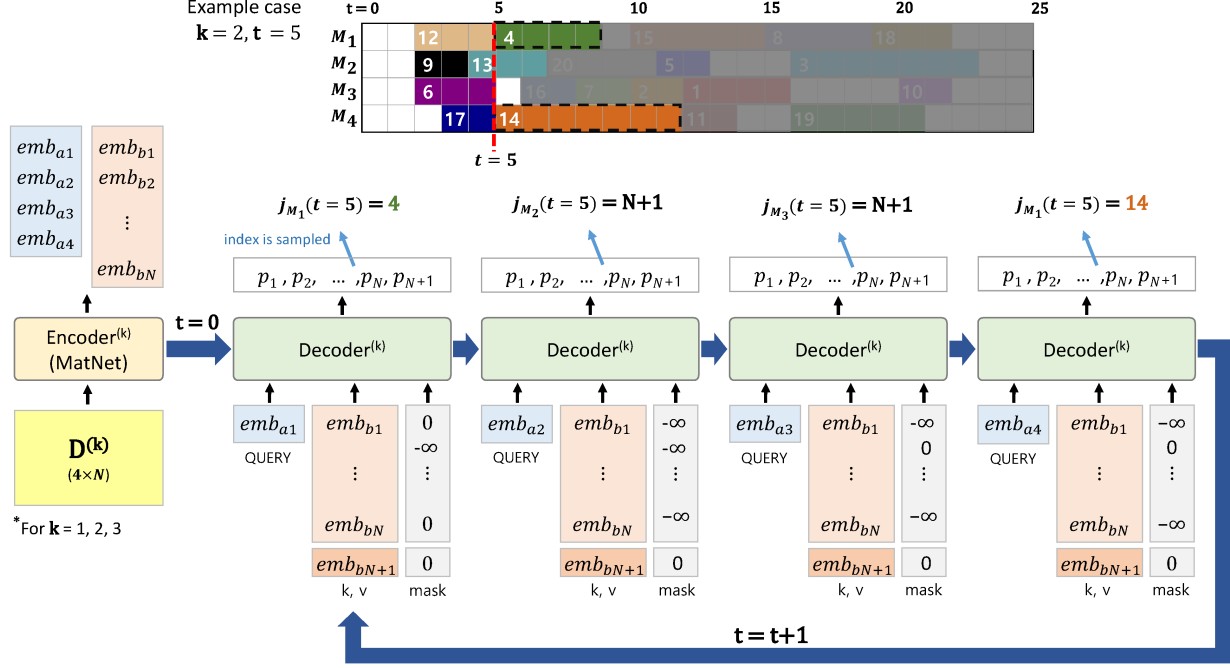

Figure J.2: FFSP solver