# OpenReview forum: "Matrix encoding networks for neural combinatorial optimization"
_NeurIPS.cc/2021/Conference — NeurIPS 2021 Poster_

### Official Review · Reviewer_2Zed · 2021-07-11

**Rating:** 6
**Confidence:** 3

**Summary:**

The paper introduces a neural net model that takes relationship matrices as input for solving combinatorial optimization problems. The proposed network can be considered as a special GNN fed by a complete bipartite graph with weighted edges. The experiment results show that their end-to-end RL framework can achieve a similar performance as LKH3 for asymmetric traveling salesman problems and significantly outperform conventional OR methods for flexible flow shop problems.

**Limitations And Societal Impact:**

Some comments:

1) For the experiment part, I would expect authors to provide more diagrams/charts to enable a more intuitive understanding of the proposed model.

2) It can be useful to employ sensitivity analysis to demonstrate the stability and scalability of the proposed framework.

3) I am looking forward to seeing the performance of this framework with other different combinatorial optimization in your future work.


**Main Review:**

Originality: The proposed model can be seen as a special GNN with a complete bipartite graph with weighted edges as input. The structure is inspired by [1] with several improvements.

1) There are two independent update functions for nodes in two disjoint sets.
2) The attention score considers the edge weight as well.

Quality: The paper is complete with sufficient details about the model structure.

Clarity: The paper is clearly written and easy to follow.

Significance: The proposed framework takes the matrix-style relationship as input, which is novel and can handle a large class of combinatorial optimization problems. The experiment results show that the proposed algorithm significantly outperforms conventional OR methods for flexible flow shop problems which are also very impressive.


[1] Wouter Kool, Herke van Hoof, and Max Welling. Attention, learn to solve routing problems! International Conference on Learning Representations, 2019.

**Time Spent Reviewing:**

10

---

> ### Author Response · Authors · 2021-08-10
> **Answers to Reviewer 2Zed**
>
>
> Thank you for reviewing our paper. We are glad to see that you have accurately assessed the significance of our work in your review. We believe our work will play as an important stepping stone for many other advanced RL methods, which will help to solve various types of practical CO problems in the future. We hope that you share your thoughts on our paper more directly so that we have chance to clarify any of your doubts, if possible. At this time, we have hard time understanding the grounds for your overall rating of our paper.
>
> &nbsp;
>
> **More diagrams/charts to explain the experiments**
>
> We realize that we could have conveyed the flow of the experiments more intuitively using visual aids. Thank you for pointing this out to us. We agree that for readers who are not familiar with the CO solution construction methods using the encoder-decoder style models, the flow of the experiments can be a little confusing. In our revision, we will include a diagram that displays how the data is processed by MatNet and then passed to the decoder, which is then used repeatedly to produce a solution sequence one item at a time.
>
> &nbsp;
>
> **Sensitivity analysis**
>
> As you are aware, MatNet is based on Transformer architecture, which is widely used in various applications and generally believed to have good stability/scalability. There are many hyperparameters used in our experiments, and none shows unusual sensitivity when they are perturbed. Is there a particular set of hyperparameters you have in mind that you would like to check? We agree that the strong stability/scalability of a model is one of the key performance features, and we welcome any idea that will help evaluate our framework and strengthen it.

---

> ### Author Response · Authors · 2021-08-23
> **Diagrams to Better Explain the Experiments**
>
> We are preparing figures that can help readers understand the flow of the experiments better.
> For the ATSP experiment, for example, you can have a look at the current version of the diagram in the following link:
>
> <https://drive.google.com/file/d/12Plcbo7CJwLbn4hC_BJZgQ6dvcXfqggh/view>
>
> Here we describe how the model generates s(𝒊), the 𝒊-th element of the solution sequence, for 𝒊=1, 2, ..., N. That is, the created ATSP solution trajectory is τ = ( s(1), s(2), …, s(N) ). The inner structures of the encoder and the decoder are already described in detail in the paper, so we omit those from the diagram for better readability.

---

### Official Review · Reviewer_i9Xj · 2021-07-14

**Rating:** 7
**Confidence:** 4

**Summary:**

This work proposes a novel matrix encoding network (MatNet) to solve combinatorial optimization (CO) problems with matrix-style relationship data. This kind of CO problems, such as asymmetric TSP, can be found in many real-world applications, but cannot be solved by the current neural combinatorial optimization approaches. The key contribution is the proposed dual graph attentional layer, along with the multi-head mixed score attention, to efficiently incorporate the matrix data into the well-known Attention Model (AM) architecture for solving CO problems.

**Limitations And Societal Impact:**

Yes.

**Main Review:**

I generally find the idea and the proposed model novel and interesting. But the lack of some important discussions on the proposed method makes me hard to give a clear acceptance for the current manuscript. I am willing to adjust my score based on the author's response.

Strengths:

+ This paper is generally well-written and easy to follow.

+ This work is the first neural combinatorial optimization approach to solve CO problems with matrix-style relation data, which is important for many real-world applications.

+ The proposed dual graph attention layer is a novel and reasonable generalization of AM model to incorporate matrix-style relation data. The design of simple initial node embedding + matrix data enhanced attention structure is novel and inspiring. The instance augmentation strategy is also an efficient way to generate diverse solutions (and to find the best one).

+ It is glad to see the proposed method can successfully solve general asymmetric TSP and flexible flow shop problems with promising results and fast runtimes.

Major Concerns:

- C1. Contribution and Better Position: The proposed dual graph attention layer and mixed score attention are novel to deal with CO problems with matrix data. However, it is not sure whether similar structures (e.g., with extra data) have been proposed in GAT/GNN/Transformer for other applications. The current related work section is mostly on neural combinatorial optimization (NCO), but a better position with other similar GAT/GNN architectures is needed to judge the novelty of the proposed architecture. Even if similar structures have been proposed, it will not hurt this work's contribution on solving CO problems with matrix data. Indeed, providing related work on similar GAT/GNN structures would be valuable for potential follow-up and improvement.

- C2. Generalization to New Problems: The generalization ability is a key feature for a learning-based solver to be used in real-world applications. Some current work finds the NCO approach is not good at generalization to problems with different distributions, which is a crucial limitation [1,2]. It seems that the proposed model makes itself harder to generalize to problems with different sizes (e.g., train on 100-city ATSP, generalization to 500-city ATSP) due to the at most $N_{max}$ one-hot initial node representation, which has been briefly discussed in the paper. Is it possible to use random 0/1 representation (total $2^N$ possibility) or even truly random representation for initial node representation to avoid this limitation? Is the one-hot vector crucial for the proposed methods?

- C3. Comparison to Other Learning-based methods: This work mainly compares the proposed method with different MIP and heuristics methods, but not other learning-based methods. It is understandable since it is the first method to deal with CO problems with matrix data. However, to better analyze the proposed model, it is still valuable to compare it with other NCO methods on problems that can be solved by both methods. For example, it is possible to use this method to solve euclidean TSP with matrix data, and compare the results obtained by POMO-based AM with classical euclidean input. Will this method perform poorer due to the lack of strong explicit prior knowledge of the euclidean coding?

- C4. Runtime: For the TSP results, MatNet takes 2s/8s/34s to solve 10,000 ATSPs with 20/50/100 cities, and POMO only needs <1s/2s/11s to solve 10,000 TSPs as reported in [3]. This resuts is reasonbale since MatNet has a larger encoder to incoporate the matrix data. However, why does the LKH3 solver only need 1s/11s/1m for 10,000 ATSPs but 42s/6m/25m for 10,000 TSPs?

Other Issues:

- I1. What does the front-end model mean in the context of NCO?

- I2. In the TSP decoder, why not also use the graph embedding (e.g., average node embedding) to make the QUERY token as in AM?

- I3. For ATSP, why the tmat class is more interesting than the general ATSP problem?

- I4. The provided Github repository is empty. I think it is unsuitable to say the code is provided in the checklist.

[1] Joshi, Chaitanya K., Quentin Cappart, Louis-Martin Rousseau, Thomas Laurent, and Xavier Bresson. Learning TSP requires rethinking generalization. arXiv preprint arXiv:2006.07054 (2020).

[2] Michal Lisicki, Arash Afkanpour, and Graham W. Taylor. Evaluating Curriculum Learning Strategies in Neural Combinatorial Optimization. arXiv preprint arXiv:2011.06188 (2020).

[3] Yeong-Dae Kwon, Jinho Choo, Byoungjip Kim, Iljoo Yoon, Youngjune Gwon, and Seungjai Min. Pomo: Policy optimization with multiple optima for reinforcement learning. Advances in Neural Information Processing Systems 33, 2020.

**Time Spent Reviewing:**

8

---

> ### Author Response · Authors · 2021-08-10
> **Answers to Reviewer i9Xj**
>
>
> Thank you for your insightful and stimulating comments! We address your concerns below:
>
> &nbsp;
>
> **C1. Related works on similar model architectures**
>
> We think that it is a great idea and we will add a separate subsection under Related Work where we provide references to other similar GAT/GNN structures or other bipartite-graph related problems found in the literature.
>
> [1] presents a classic GNN encoding scheme applied on a bipartite graph, which is used to learn effective branch-and-bound variable selection policies for solving MIP instances. The two groups of nodes represent constraints and variables of an MIP instance and their links are weighted by constraint coefficients. [2] tries to find good auction policies given a matrix that describes the value that each bidder assigns to each auctioning item. [3] solves weapon-target assignment problem, a classic CO problem that is presented with a bipartite graph. [4] deals with a sparse, large scale (>billions of nodes) bipartite graph. Graph convolution via random walks is introduced that creates node representations useful for grouping or quickly finding related nodes for recommendation. Papers by [5] and [6] are not about bipartite graphs, but one can find graph embedding schemes that use both the graph attentional layers and the edge weight mixing, somewhat similar to our methods.
>
> MatNet operates on a bipartite graph that has no node features and has weighted edges. Note that none of the previous works above have considered such a case and cannot serve as good substitutes for MatNet to solve the CO problems presented in our paper.
>
> &nbsp;
>
> **C2. One-hot initial node representations**
>
> You have raised a very good point. For initial node representations (of nodes in group B), one only needs mutually distinct-enough N_max vectors and they do not need to be of one-hot type. We have chosen to present our model with one-hot vectors because it is the simplest to implement this way. But for the readers who look for more generalizability, this may seem too restrictive.
>
> In our revision, we will add more discussion on the topic of the initial vector preparation and provide more experiment results. The suggested “random 0/1 representations” would work as replacements for one-hot vectors, but our favorite choice throughout the development stage of MatNet has been a set of trainable vectors, which automatically becomes well-performing (as good as one-hots), mutually-distinct vectors during the model training. This scheme has an advantage that one has precise control on what N_max value to use to accommodate for possible future generalization applications.
>
> Of course, having too large N_max comes with a price. For example, one could use random vector representations, which completely lift the restriction of N_max. But our experiment with ATSP50 shows that when we switch one-hot vectors to random sequence vectors (both of length 50), the solver’s performance on ATSP50 drops from the average tour length of 1.5801 (optimality gap 1.34%) to 1.5878 (1.83%) without the instance augmentation.
>
> Having a large N_max, on the other hand, means there are greater varieties of initial node embeddings one can take advantage of so that the instance augmentation scheme becomes even more powerful. With a 128x instance augmentation, the gap between the one-hot encoding scheme and the random one quickly becomes smaller: 1.5610 (0.11%) vs. 1.5640 (0.30%). With a 1280x instance augmentation, it becomes 1.5598 (0.03%) vs. 1.5615 (0.14%).
>
> Lastly, we would like to point out that Appendix F in our supplementary document describes a generalization experiment using FFSP. Here, we also explain why MatNet can be freely generalized, even with the tight restriction of N_max, in many areas of practical applications where the data matrices change in length only in one dimension.
>
> &nbsp;
>
> **C3. Comparison with (POMO-applied) AM on Euclidean TSP**
>
> We agree that comparisons with other neural CO methods using common problems can help the readers to better assess MatNet. In our revision, we will include the results and the analysis of the experiments that compare the performance of our proposed MatNet-based solver with that of POMO-AM on the same Euclidean TSPs. The preliminary result on TSP50 shows that MatNet-based solver performs better than POMO-AM. This came as a surprise to us, because MatNet is structurally designed to handle much broader range of problems while the AM specializes in solving the symmetric ones (i.e., the AM has the hard-coded prior knowledge of the problem instances). It would be interesting to see the results on TSP100, for which the performance of the two models can be contrasted more clearly. For TSP50, both methods produce solutions that are already too close to the optimal (optimality gap being less than 0.03%) so that we need more careful analysis before presenting the numerical results.
>
> &nbsp;
>
> **C4. LKH3 runtimes**
>
> You have correctly noticed that the MatNet-based solver needs more time than its counterpart AM during the inference stage because it needs to encode the matrix-type data which is computationally more expensive. As for LKH3, our way of measuring the runtime differs from that of [7] in two major points, both of which are described in the text. (1) We exclude the program loading time and the data (matrices) loading time from the storage, and only accumulate the real computation time of the algorithm. (2) We divide the runtime of LKH3 by a factor of 8, just to provide a little more balanced point of view between the CPU- and the GPU-based processes. This is because LKH3 can be run in parallel on multi-core processors using multi-threading without too much trouble. (This is not true for CPLEX, which already uses multi-threading internally.) Of course, this factor 8 is somewhat arbitrary. We have commented in the text that this should not be taken too seriously.
>
> &nbsp;
>
> **I1. Meaning of “front-end model”**
>
> Front-end model takes the data (parameters) explicitly defined by the problem instance and processes it to extract global information at the scope of the instance. A back-end model would then use the processed information passed by the front-end model and produce a solution. For an end-to-end neural solver, the distinction between the two blurs, but in our paper we generally meant the encoder module (i.e., MatNet) when we used the term “front-end model”. In hybrid approaches such as ML-MIP and ML-heuristics where the front-end model is a neural net and the back-end model is not, the distinction is clearer.
>
> &nbsp;
>
> **I2. Graph embedding inside the QUERY token**
>
> During our ablation studies, we found that the averaged node embeddings included in the QUERY token makes a negligible effect on the solver’s performance. So we left out the graph embedding info from the QUERY token (for clarity of the model, perhaps). Note that this is not just the case for our ATSP solver. We have found that the graph embedding can be omitted safely from the QUERY used in the AM-based TSP solvers such as those in [8] and [7].
>
> &nbsp;
>
> **I3. Tmat class ATSP**
>
> Imagine that it takes 10 hours to go from City-A to City-B directly, but only 1 hour to travel from City-A to City-C and also 1 hour from City-C to City-B. It makes no sense to impose a restriction that we cannot travel from A to B in 2 hours by going through C, especially when we go into all this trouble to find the fastest traveling route when we solve for TSPs. Routing problems without the guaranteed triangle inequality are purely theoretic and has little practical value.
>
> &nbsp;
>
> **I4. Github repository**
>
> The codes were uploaded on July 12th. We meant to have it up before the review started, but it seems like we have missed it by a few days. We apologize for the confusion. It is an anonymized repository, meant only for the reviewers. The codes will be moved to a permanent location when the review period is over.
>
> &nbsp;
>
> [1] Gasse, Maxime, et al. “Exact Combinatorial Optimization with Graph Convolutional Neural Networks.” Advances in Neural Information Processing Systems, vol. 32 (2019).
>
> [2] Duetting, Paul, et al. “Optimal Auctions through Deep Learning.” 36th International Conference on Machine Learning (2019).
>
> [3] Gibbons, Daniel, et al. “Deep Learning for Bipartite Assignment Problems.” 2019 IEEE International Conference on Systems, Man and Cybernetics (2019).
>
> [4] Ying, Rex, et al. “Graph Convolutional Neural Networks for Web-Scale Recommender Systems.” Proceedings of the 24th ACM SIGKDD International Conference on Knowledge Discovery & Data Mining (2018).
>
> [5] Sykora, Quinlan, et al. “Multi-Agent Routing Value Iteration Network.” ICML 2020: 37th International Conference on Machine Learning (2020).
>
> [6] Dwivedi, Vijay Prakash, and Xavier Bresson. “A Generalization of Transformer Networks to Graphs.” ArXiv Preprint ArXiv:2012.09699 (2020).
>
> [7] Kwon, Yeong-Dae, et al. “POMO: Policy Optimization with Multiple Optima for Reinforcement Learning.” Advances in Neural Information Processing Systems, vol. 33 (2020).
>
> [8] Kool, Wouter, et al. “Attention, Learn to Solve Routing Problems!” International Conference on Learning Representations (2019).

---

> > ### Comment · Reviewer_i9Xj · 2021-08-13
> > **Follow-up Questions**
> >
> > Thank you for your detailed and to-the-point response to my concerns, where many of them are properly addressed. I have a few follow-up questions:
> >
> > + **C2. One-hot initial node representation:** It is glad to know the generalization limitation can be overcome by replacing the one-hot vectors. I am a bit curious about where the price comes from when you switch to the random sequence vectors. Why can one-hot vectors have better performance than random vectors (how the similarity among node representation affects the model performance)? It is also interesting to know the details of the trainable vector approach.
> >
> > + **C3. Euclidean TSP:** It is quite surprising to find MatNet can outperform POMO-AM on Euclidean TSP. Would it be caused by different model sizes? How about let POMO-AM have a similar size with MatNet (more attention layers or larger attention embedding)? More experiments and proper discussion would be crucial to validate and analyze this result.
> >
> > + **I2. Graph embedding:** It is also a surprise to me that the graph embedding can be fully dropped from the QUERY token for ATSP as well as other AM-based TSP solvers. Does it mean the decoder can only depend on the embedding of the first node and the current node? In this case, it seems the information of other nodes (and the graph) are only aggregated into the first and current nodes via the attention scheme in the encoder, which is relatively weak and mostly represents the local graph structure among the two nodes.

---

> > > ### Author Response · Authors · 2021-08-18
> > > **Thank you for your follow-up questions.**
> > >
> > >
> > > **C2-1. On similarities among initial node representations:**   Assume we have three nodes, A, B, and C, and let’s further assume that A-B has much stronger relationship than A-C. Roughly speaking, a single layer of the graph attention network will update the representation of node A by mainly adding the representation of B (or more specifically, the “value (v)” part among the “qkv" of B) to that of A. The representation of C will also be added to that of A, but in a much smaller scale, because of its weaker relationship with A (i.e., smaller “score” in the attention mechanism). When the initial representations of B and C are equal (an extreme case of being similar, for the simplicity of the argument), the updated representation of A cannot provide any information as to whether it is B or C that has stronger relationship with A.
> > >
> > > With more and more attentional layers stacked together, the representations of B and C will become increasingly distinct and the non-distinguishability effect explained above will become less prominent. But hopefully our hand-wavy explanation above can provide some theoretical insight on the importance of non-similarity of the initial node representations.
> > >
> > > On the empirical side, we have done another follow-up ATSP50 experiment. In addition to ATSP50 experiment with random vectors of length 50 (explained in our previous response to your initial review), we have performed the same experiment with random vectors of length 20. Having shorter length forces initial vectors to be more similar to each other. This change (50 -> 20) makes the training slower and degrades the final performance as well. We will add this result to Appendix too..
> > >
> > > &nbsp;
> > >
> > > **C2-2. Trainable vectors for initial node representations:**   The use of trainable vectors in these types of models is not new. In the TSP solver of Kool et al. [1], a trainable vector is used to make “START” token for the decoder. Our approach is similar. We prepare N_max number of trainable vectors filled with random numbers and use them as initial node representations. During the training stage, the values of these vectors are updated, just like the parameters of the neural networks, via reinforcement learning. At the inference stage, we use the final values of these trainable vectors. When the length of these trainable vectors is N_max or more (i.e., the length of the vectors is larger than or equal to the number of the vectors), this is effectively equivalent to one-hot vector initial node representation scheme described in our paper, as they will be trained to be mutually orthogonal.
> > >
> > > &nbsp;
> > >
> > > **C3. Euclidean TSP:**   We are working on designing and running the comparison experiments using TSP100 in a more careful way (similar model size) as you have suggested. In the meantime, we want to point out that one of the main advantage the MatNet-based model has over the AM, observed from our preliminary tests (based on TSP50), is its ability to use the instance augmentation freely. Without the instance augmentation, MatNet-based model slightly underperforms POMO-AM.
> > >
> > > However, we have been using 128x instance augmentation for MatNet by default. POMO-AM, on the other hand, can only use 8x augmentation at the maximum. In order to make a fair comparison, we should force POMO-AM to produce even more diverse trajectories beyond 8x instance augmentations. But this is only possible if we switch the inference scheme of POMO-AM from greedy (based on “argmax” selection) to probabilistic sampling. This, unfortunately, makes the performance of POMO-AM drop significantly.
> > >
> > > All in all, the comparison between the two models on Euclidean TSP is interesting, but our major takeaway message should be that the MatNet-based model shows more or less the similar performance of POMO-AM. Detailed descriptions of which one showing slightly better performance over the other in what settings, as stated above, seem not too insightful on these models.
> > >
> > > &nbsp;
> > >
> > > **I2. Graph embedding:**    The attention mechanism makes a weighted sum of the “values.” If the weights are made the same (via constant values used in creating “keys” and “queries”) and if the “values” are just the copies of the representations of the nodes (via identity transformation), the output of the attentional layer is the sum of the representations of all the nodes, which is the “graph embedding.” The point is, it is possible to contain the “graph embedding” information in the embedding of any node, when the reinforcement learning process finds it necessary. Dropping “graph embedding” from the QUERY does not necessarily mean that the view of the solver is now restricted to a local one, or any less than that of the solver equipped with a QUERY containing “graph embedding.” The empirical evidence supports this.
> > >
> > > &nbsp;
> > >
> > > [1] Kool, Wouter, et al. “Attention, Learn to Solve Routing Problems!” International Conference on Learning Representations, 2019.

---

> > > > ### Comment · Reviewer_i9Xj · 2021-08-20
> > > > **Thank you for the follow-up response**
> > > >
> > > > Thank you for the detailed follow-up response. I increase my score to 7 and have no further comment at this moment. It is important to add the discussions and new experimental results in the paper (or the appendix).

---

### Official Review · Reviewer_tewG · 2021-07-14

**Rating:** 5
**Confidence:** 5

**Summary:**

The authors propose a family of matrix encoding networks to encode the states of combinatorial optimization problems which can be encoded as a matrix. The proposed matrix encoding network works like a bi-partite graph embedding network. Compared to previous efforts encoding sequences and graphs, the effort of this paper is appealing. Two CO problems are studied: ATSP and FFSP, and the performance on ATSP is inferior to competing methods, and the performance on FFSP is more convincing.

**Limitations And Societal Impact:**

The limitation part is clearly written and well discussed. I agree with the authors:

“A neural heuristic solver that clearly outperforms conventional OR methods has been rare, especially for classical optimization problems. Perhaps, this is because the range of CO problems ML researchers try to tackle has been too narrow, only around simple problems (such as the ATSP) for which good heuristics and powerful MIP models already exist.”


**Main Review:**

The strengths of this paper are:
1. Designing encoding networks for matrices in neural CO is interesting and appealing, deferring from previous efforts in modeling sequence and graph data.
1. The designed attention-based network is similar to the Transformer architecture, and the unsymmetric message-passing design based on the bi-partite graph is reasonable.
1. Experiment results on FFSP are convincing.
1. The authors discuss the potential of applying the proposed method to hybrid ML-OR approaches.

The limitations of this paper are:
1. My major concern is that the experiment results in this paper may not ground this paper as a general framework for a wide range of CO problems that can be modeled by matrices. Because the authors only experiment on two CO problems (ATSP and FFSP), and only the FFSP results are convincing. If this paper is focused on FFSP, I think it is a good submission; however, since the authors are aiming at a wider range of problems, the experiment results are not that convincing. In comparison, most existing papers which focused on developing a general approach (e.g. S2V-DQN in NeurIPS17, NeuRewritter in NeurIPS19) study 3 different CO problems and report convincing results on all problems.
1. Since MatNet uses zero-vectors to embed nodes in A, and one-hot vectors for nodes in B, can we design a MatNet for symmetric cases (e.g. symmetric non-Euclidean TSP)?

Some minor remarks:
1. As a read from "Instance augmentation via different sets of embeddings", do you break the permutation invariance nature of MatNet?
1. In Table 2, Random and Shortest Job First are slower than MatNet. However, from my understanding, Shortest Job First is a greedy algorithm and should be at least as fast as MatNet (which performs greedy roll-out), because Shortest Job First does not require the forward pass of neural networks. Random should also be a fast algorithm for the same reason.

**Time Spent Reviewing:**

6

---

> ### Author Response · Authors · 2021-08-10
> **Answers to Reviewer tewG**
>
>
> Thank you for reviewing our paper and finding our effort appealing. Please let us address your concerns below.
>
> &nbsp;
>
> **[Limitation 1-1] Not convincing enough result: Performance comparison with existing methods**
>
> Your statement that most existing deep RL papers developing general CO approaches have reported results that are superior to the competing methods is incorrect. In fact, the exact opposite is true. While any key neural CO paper would do, let us review the two RL papers you have mentioned as examples.
>
>
> **<S2V-DQN [1]>**  In this paper, (1) MVC and (2) MAXCUT problems are solved using CPLEX to find the close-to-optimal solutions. (3) TSP is solved using Concorde algorithm. The solutions of these existing methods are compared to the ones calculated by S2V-DQN. The reported “approximation ratios” are larger than 1 in all cases, meaning that solutions from S2V-DQN are inferior to both CPLEX and Concorde in all three types of problems.
>
> **<NeuRewriter [2]>** (1) Halide language expression simplification problem: This is a highly domain-specific problem (i.e., not a typical, classic CO problem). In this problem, we admit NeuRewriter did perform the best among the other problem-specific methods. (2) Job scheduling problem: NeuRewriter performed the best, but only against simple heuristics such as SJF (shortest job first). Unlike our paper, no meta-heuristic approaches (such as genetic algorithms) have been tested. Meta-heuristics are generally the ‘to-go’ methods for these types of problems and guaranteed to perform better than SJF. (3) CVRP: once again, good existing methods have not been included in the list of baselines in their paper. Gurobi (an MIP solver similar to CPLEX) is mentioned in their appendix, compared to which NeuRewriter performed worse. Also, we note that the reported performance of NeuRewriter is far inferior to that of the LKH algorithm reported in the preceding papers (e.g., the LKH results reported in the AM paper (Kool et al. 2019 [3]), which NeuRewriter paper cites).
>
>
> As we have stated in our paper, neural CO solvers outperforming conventional OR methods are extremely rare. But the field of neural CO has just started, compared to the long history of OR research. Every year new publications come out and bring the gap between the two a little closer. Less than 1% optimality gap we have achieved in ATSP100 (solved by deep RL for the first time) is a spectacular result, by any measures in the field. When TSP100 was solved using deep RL for the first time (Bello et al. 2017 [4]) the optimality gap was 7.8% without sampling, and 1.7% with 1,280,000 samplings plus 10k extra training steps at the inference stage (active search). Additionally, S2V-DQN [1] solved TSP100 with the optimality gap of 7-8%.
>
> And as for our FFSP results, our work is actually one of the very first few demonstrations of the purely neural end-to-end approaches that have successfully beat the conventional OR methods in solving a classic CO problem.
> Our work marks an important stepping stone, just like other previous key neural CO papers. And this is not because these methods have outperformed many existing methods. We have laid out a foundation work, providing a general neural approach for CO problems that have not been previously attempted, so that the community can keep expanding its area of interest and continue to build more advanced techniques.
>
> &nbsp;
>
> **[Limitation 1-2] Not enough number of example problems: Demonstration of two problems, instead of three**
>
> We have chosen two problems of vastly different nature and have demonstrated exceptional results on both. There is no shortage of CO problems that are defined upon matrix-type parameters, both theoretical and real-world-based. And how one applies MatNet to encode these matrix-style relationship data does not depend on the specifics of the problems. We have focused our attention in showing step-by-step, in-depth analyses of the performed experiments. Given the tight space restriction of NeurIPS, one needs to make a decision on what is best for the readers.
>
> &nbsp;
>
> **[Limitation 2] MatNet on symmetric problems**
>
> Whether symmetric or asymmetric, MatNet can encode any given matrix. For example, following the discussion with Reviewer i9Xj - C3, we are running experiments on symmetric, Euclidean TSP using MatNet, the result of which will be added in our revision. The preliminary result shows that MatNet-based solver is already outperforming the current SOTA method (POMO-AM [5]).
> A symmetric variant of MatNet that you suggest, however, would certainly improve in performance even more when dealing with such symmetric problems. We believe it is best left for a topic of further research, which is a very interesting one.
>
> &nbsp;
>
> **[Minor 1] Permutation invariance nature of MatNet**
>
> You have correctly observed that the assignment of different one-hot vectors to nodes in B hinders MatNet from being completely permutation-invariant. Fortunately, this has enabled the extensive use of instance augmentation technique that boosts the performance of MatNet-based CO solvers. Nevertheless, a completely permutation-invariant model for matrix encoding is still an open problem, an important research topic for us in the future.
>
> &nbsp;
>
> **[Minor 2] Runtimes for Random and SJF**
>
> The reported runtimes for Random and SJF in Table 2 are slower than those of MatNet because these heuristic algorithms are programmed to solve problems in a sequential fashion, as any regular, conventional CO algorithm would do. MatNet, on the other hand, solves problems in mini-batches in parallel using GPUs, as is customary for neural approaches. But your understanding is correct and if we modify our Pytorch-based MatNet code to implement Random or SJF and make them solve batches of FFSPs in parallel, it would run much faster than MatNet.
>
> (*Side note*) Random algorithm is computationally lighter than SJF and should run faster than SJF. But SJF has shorter runtime in Table 2! This is because SJF solves FFSP better and therefore its episodes finish earlier than those of Random. (i.e., the Gantt charts SJF produces are shorter in length, so it makes SJF seem to run faster.)
>
> &nbsp;
>
> [1] Dai, Hanjun, et al. “Learning Combinatorial Optimization Algorithms over Graphs.” NIPS’17 Proceedings of the 31st International Conference on Neural Information Processing Systems, vol. 30, 2017.
>
> [2] Chen, Xinyun, and Yuandong Tian. “Learning to Perform Local Rewriting for Combinatorial Optimization.” Advances in Neural Information Processing Systems, vol. 32, 2019.
>
> [3] Kool, Wouter, et al. “Attention, Learn to Solve Routing Problems!” International Conference on Learning Representations, 2019.
>
> [4] Bello, Irwan, et al. “Neural Combinatorial Optimization with Reinforcement Learning.” ICLR (Workshop), 2017.
>
> [5] Kwon, Yeong-Dae, et al. “POMO: Policy Optimization with Multiple Optima for Reinforcement Learning.” Advances in Neural Information Processing Systems, vol. 33, 2020.

---

> > ### Comment · Reviewer_tewG · 2021-08-11
> > **I will raise my rating by 1 point after reading the rebuttal**
> >
> > Many thanks for the detailed response from the authors addressing my questions. I will raise my rating by 1 point (4->5) after reading the rebuttal. However, I have the following questions that seem unclear to me:
> >
> > 1. The authors describe the proposed framework as a general for matrix-encoded CO problems, and it does not depend on the specifics of the problems. In line 41 I am aware that the authors describe that their MatNet is capable of handling quadratic assignment problems. Can the authors provide further feedback about how to design MatNet for quadratic assignment problems?
> >
> >    From my knowledge, the number of elements of the matrix in the quadratic assignment problem is of the size of $n^2m^2$, where $n, m$ denotes the number of asymmetric items considered in the problem. For symmetric problems, there are $m=n$. In contrast, the number of elements of matrices in the linear assignment problem, the job-shop scheduling problem are with the size of $nm$. From my own perspective, the proposed framework may not generalize so easily to "special" problems like the quadratic assignment problem.
> >
> > 2. Your effort in providing the symmetric TSP result is appreciated, however, can you list the detailed results with comparison to learning/non-learning baselines in a table? Openreview supports markdown formatting and also supports inserting tables.

---

> > > ### Author Response · Authors · 2021-08-18
> > > **Thank you for your follow-up questions.**
> > >
> > >
> > > **1. Quadratic Assignment Problem:**  (We aren’t certain about your description on quadratic assignment problem.) One definition of quadratic assignment problem (QAP) can be found in: <https://en.wikipedia.org/wiki/Quadratic_assignment_problem>. In this case, parameters that define a problem instance is two N-by-N matrices. MatNet can be used to encode each of these matrices.
> > >
> > > Once again, we would like to emphasize that the goal of our paper is to present MatNet, which encodes matrices and produces good representations for the entities of the problem. The design of a decoder depends on the type of a problem. The two end-to-end neural solvers in our paper exemplify how MatNet and the existing neural decoder structures can be combined. The decoder, however, can be implemented by a neural net, or heuristically, or via MIP. It is up to you who can perform experiments to find the most effective one.
> > >
> > > On a side note, it is not too difficult to come up with a simple decoder design for QAP. One crude design idea is the following. A decoder will sort the “facilities” and the “locations” of QAP based on their representations returned by MatNet. The facility and the location of the same ranking can be matched, which will create a solution (made of N pairs) required by the problem. There are better ways to construct a decoder, but we will reserve these as a topic for another paper.
> > >
> > > &nbsp;
> > >
> > > **2. Symmetric TSP:**  If you are interested in seeing the comparison of recent results of learning/non-learning methods on symmetric (Euclidean) TSP, we refer you to the result tables of Kwon et al. [1] and Kool et al. [2]. The performance of our MatNet-based solver is not too different from those of (AM-based) POMO [1]. We are working on experiments that compare the two (MatNet and the AM) more carefully for the camera-ready version, but what is most important here is that the performance of the two are comparable.
> > >
> > > &nbsp;
> > >
> > > [1] Kwon, Yeong-Dae et al. “POMO: Policy Optimization with Multiple Optima for Reinforcement Learning.” Advances in Neural Information Processing Systems, vol. 33, 2020.
> > >
> > > [2] Kool, Wouter et al. “Attention, Learn to Solve Routing Problems!” International Conference on Learning Representations, 2019.

---

> > > > ### Comment · Reviewer_tewG · 2021-08-18
> > > > **Thank you for the response and here are my follow-up questions**
> > > >
> > > > 1. About the quadratic assignment problem, I might be more interested in the so-called Lawler's Quadratic Assignment Problem (Lawler's QAP), which is the most general form of QAP and there is only one matrix with $m^2n^2$ elements in Lawler's QAP. The formulation in your mentioned wiki page is known as Koopmans-Beckmann's QAP, which can be viewed as a special case of Lawler's QAP. Perhaps you can refer to the original Lawler's paper instead of Wikipedia: https://www.jstor.org/stable/2627364
> > > >
> > > >    Anyway, I agree the quadratic assignment problem may be beyond the scope of this paper, I am just happy to see the response from the authors about the claims made in the paper that may seem unclear to me.
> > > >
> > > > 2. About the Symmetric TSP, I am a little bit confused why the authors cannot just post their preliminary results on TSP50 as a table. Of course, if your further experiment achieves better results, you can update the results in the camera-ready version.

---

> > > > > ### Author Response · Authors · 2021-08-20
> > > > > **Preliminary results on Euclidean TSP50**
> > > > >
> > > > > Our preliminary result is based on a MatNet model (=the same model used in the paper) that has a somewhat larger size than POMO-AM [4]. A fair comparison may result MatNet-based solver having a slightly worse performance (but faster runtime). The following table is a copy from [4] plus our results. It shows the performance of each non-learning-based and learning-based method on solving 10,000 randomly generated TSP (50 nodes are selected from a unit square by uniform random). The displayed gap is the optimality gap, where the optimal solutions come from Concorde algorithm on the same problems.
> > > > >
> > > > > Lastly, we would like to point out that the instance augmentation technique on the POMO-AM method is limited to 8x, and larger augmentations (such as 128x like ours) are not possible.
> > > > >
> > > > > &nbsp;
> > > > >
> > > > > | Method | Length |  Gap   | Time |
> > > > > | -----------    | -----------: | ---: | ---: |
> > > > > | Concorde | 5.69      | - | (13m) |
> > > > > | LKH3        | 5.69     | 0.00% | (6m) |
> > > > > | Gurobi      | 5.69      | 0.00% | (2m) |
> > > > > | Farthest Insertion      | 6.00      | 5.53% | (2s) |
> > > > > | GCN [1], +beam search      | 5.69      | 0.01% | (18m) |
> > > > > | Improvement [2], 5000 iter.    | 5.70      | 0.20% | (1h) |
> > > > > | Improvement [3], 2000 iter.    | 5.70      | 0.12% | (29m) |
> > > > > | POMO+AM [4], no aug.    | 5.70      | 0.21% | (2s) |
> > > > > | POMO+AM [4], 8x aug.    | 5.69      | 0.03% | (16s) |
> > > > > | OURS, no aug.                 | 5.71      | 0.30% | (8s) |
> > > > > | OURS, 8x aug.                 | 5.69      | 0.05% | (1m) |
> > > > > | OURS, 128x aug.             | 5.69      | 0.01% | (16m) |
> > > > >
> > > > > &nbsp;
> > > > >
> > > > > [1] Joshi, Chaitanya K., et al. “An Efficient Graph Convolutional Network Technique for the Travelling Salesman Problem.” ArXiv Preprint ArXiv:1906.01227, 2019.
> > > > >
> > > > > [2] Wu, Yaoxin, et al. “Learning Improvement Heuristics for Solving Routing Problems..” IEEE Transactions on Neural Networks, 2021.
> > > > >
> > > > > [3] Costa, Paulo R.de O. da, et al. “Learning 2-Opt Heuristics for the Traveling Salesman Problem via Deep Reinforcement Learning.” ACML, 2020.
> > > > >
> > > > > [4] Kwon, Yeong-Dae, et al. “POMO: Policy Optimization with Multiple Optima for Reinforcement Learning.” Advances in Neural Information Processing Systems, vol. 33, 2020.

---

### Official Review · Reviewer_cRFQ · 2021-07-17

**Rating:** 5
**Confidence:** 3

**Summary:**

Several combinatorial optimization problems can be fully specified by one or more matrices specifying the relationships between two groups of parameters. For example, such a matrix can encode inter-city distances in the traveling salesman problem. In the hybrid flow shop problem, a set of matrices D_i encode the time it takes to perform a job on a given machine at each timestep i. the This paper proposes to learn to embed such matrices using a neural network derived from the transformer architecture and use such embedding to find good approximate solutions to the corresponding combinatorial problems.

**Limitations And Societal Impact:**

The authors appear to have been candid about the limitation of their approach.

**Main Review:**

The paper makes 3 contributions:
 * It introduces what it calls a matrix encoding network (MatNet). It details how to modify the transformer architecture to turn it into a supercharged graph attention network. This is the most significant contribution of this paper.
 * It demonstrates step by step how to build the matrices for two important combinatorial problems, the asymmetric traveling salesman problem (ATSP) and the flexible flow shop problem (FFSP). The paper states precisely what hyper parameters were used for the magnet model, and how to derive a solution from the MatNet embedding. Providing such a detailed description of the process ensures that the results of this paper will be easily replicable.
 * It evaluates the effectiveness of the approach on the ATSP and the FFSP problems by comparing the results against a commercial grade MILP solver (CPLEX) as well as 4 heuristics based solvers.

The paper is clearly written and well organized. This makes it easy to follow. It also provides an extensive list of related work to help put the work in perspective.

The paper states that the ATSP the FFSP problems were previously not amenable to DNN based optimization. While it is possible, using previous work such as Kool et al mentioned in the paper, to encode a M by N matrix by simply embedding N features for each of the M nodes, this encoding wouldn't be flexible since the number of features would depend on one of the dimensions of the problem. As a result, a new policy would need to be trained for each value of N that is being considered. In contrast, this paper realizes that a M by M matrix can be interpreted as a bipartite graph and that a single bipartite graph neural network can, in theory, be trained to handle all matrix sizes. However, the paper appears to train distinct neural networks for each problem size, so it fails to demonstrate the flexibility of the MatNet embedding.

The paper explains how to adapt a transformer based neural architecture to apply it to a bipartite graph, and provides a rationale for the modifications, in particular the need for 2 update functions. However it doesn't validate these design choices, for example with an ablation study. Furthermore, it doesn't compare the effectiveness of its MatNet architecture against other bipartite graph embeddings. The work of gasse et al (Exact Combinatorial Optimization with Graph Convolutional Neural Networks) would be a good candidate here, but other bipartite graph embeddings solutions have been published and could be used as well.

Furthermore, RL driven MILP solvers are readily available (see for example http://ecole.ai) for academic work and could have been used in addition to CPLEX and the heuristic based approaches to build more relevant baselines for evaluation.

I believe that the MatNet approach has potential, but it is currently insufficiently evaluated.



**Time Spent Reviewing:**

6

---

> ### Author Response · Authors · 2021-08-10
> **Answers to Reviewer cRFQ**
>
> Thank you for reviewing our paper and appreciating the potential of our work! Please let us address your concerns below.
>
>
> &nbsp;
>
>
>
> **Flexibility towards different problem sizes**
>
> We disagree with your statement that we have “failed to demonstrate the flexibility of the MatNet embedding” because we have “trained distinct neural networks for each problem size.” Our experiments use one neural net (architecture) that remains the same for different problem sizes (N=20, 50, 100). This way of demonstrating flexibility (i.e., separately training the same neural model for each problem size to make distinct domain-specific solvers) has been a long tradition in the publications of the neural CO community, which we have faithfully followed. This evaluation scheme can be seen from Bello et al. (2016) [1] to Kool et al. (2019) [2] and all the other similar publications.
>
> Perhaps you were expecting to see a model that is trained to solve problems of all three domains (N=20, 50, and 100) at the same time. Such demonstration may be useful, but generally considered not necessary, because it can be simply done by adopting a training procedure that uses a mixture of N=20, 50, and 100 problem instances as the training data. Note that no previous key papers in this field, such as those mentioned above, have had the need to carry out such experiments to prove the flexibility of their models.
> A related topic is a generalizability of a trained model. That is, when a model is trained only for N=100 problems, can it also solve problems of the kind that it is not prepared for (e.g. N=20, 50, or even 1000) reasonably well? For interested readers, we describe such an experiment using the FFSP as an example in Appendix Section F.
>
>
> &nbsp;
>
>
>
> **Comparison with other bipartite graph embeddings schemes**
>
> There are some existing publications that use bipartite graph embeddings, but no previous work supports embedding of a single matrix. The bipartite graph that represent a matrix, described in our paper, has many particular characteristics. It is a complete, undirected graph, and all the edges are weighted by real numbers while the nodes themselves do not possess any features. Gasse et al. [3] present a network that encodes a bipartite graph, but it heavily relies on long list of initial node features. Blindly applying zero and/or one-hot vectors as the node features to their model leads to ineffective embeddings. Similarly, works on deep learning for assignment- or matching- type problems have devised a type of bipartite graph embeddings of their own, but they do not support null node features [4,5] or edge weights [6]. We will add a subsection under Related Work in our revised paper to make it more clear how our embedding scheme is unique in its kind.
>
> As for the ablation study on 2 update functions, we will perform and add the ablation study results in the appendix, where we already have some additional descriptions of MatNet variant structures (Section A) and other ablation tests (Section E).
>
>
> &nbsp;
>
> **RL driven MILP solvers as baselines**
>
> We would like to keep the scope of our paper on comparing MatNet results to those of more general and traditional methods. Our goal in this paper is not to win races, but rather to introduce MatNet as a new ML tool to the neural CO community. Furthermore, for the purpose of evaluating MatNet, performance variations of different MILP solvers are insignificant because MILP works too well for ATSP while just the opposite for FFSP.
>
>
> &nbsp;
>
>
>
> [1] Bello, Irwan, et al. “Neural Combinatorial Optimization with Reinforcement Learning.” ICLR (Workshop), 2017.
>
> [2] Kool, Wouter, et al. “Attention, Learn to Solve Routing Problems!” International Conference on Learning Representations, 2019.
>
> [3] Gasse, Maxime, et al. “Exact Combinatorial Optimization with Graph Convolutional Neural Networks.” Advances in Neural Information Processing Systems, vol. 32, 2019.
>
> [4] Duetting, Paul, et al. “Optimal Auctions through Deep Learning.” 36th International Conference on Machine Learning, 2019.
>
> [5] Gibbons, Daniel, et al. “Deep Learning for Bipartite Assignment Problems.” 2019 IEEE International Conference on Systems, Man and Cybernetics, 2019.
>
> [6] Ying, Rex, et al. “Graph Convolutional Neural Networks for Web-Scale Recommender Systems.” Proceedings of the 24th ACM SIGKDD International Conference on Knowledge Discovery & Data Mining, 2018.

---

### Author Response · Authors · 2021-09-01
**Planned changes for the camera-ready version**



We thank the reviewers, once again, for their time and many helpful comments. Here we summarize the planned changes of our manuscript (main text + Appendix) for the camera-ready version that we are working on.



1. A subsection describing related works that have used bipartite graph encoding or have solved assignment-type CO problems. This will show more clearly how our model/approach is unique and where we stand.



2. In-depth discussion on the topic of the initial node representations. We will present the experiment results based on the alternative representation schemes. They show slightly lower performance but have better generalization ability.



3. Ablation studies on the use of 2 update functions, using ATSP, FFSP, and Euclidean TSP.



4. Figures that explain the flow of the experiments described in the paper. This will help readers understand more intuitively how the encoder (MatNet) and the decoder are used to create CO problem solutions.



5. **Euclidean TSP experiments.** Although MatNet is intended to solve much broader types of TSP (which no previous neural method has been able to solve), showing the performance of MatNet on this special case of TSP will help the readers compare MatNet with other neural CO methods correctly. (i.e., MatNet shows superior performances.)



6. We will reflect the various comments made during the reviewing process. We will add more descriptions to the text or edit each part to be more clear.

---

### Decision · Program_Chairs · 2021-09-27

**Decision:**

Accept (Poster)

**Comment:**

The paper presents a general approach to solving approximately with GNNs a large class of matrix-based combinatorial optimization problems. After the discussion, the paper is
* an interesting contribution for combinatorial optimization problem solving from the machine learning perspective.
* will be helpful for other problems not covered in the paper.
* the paper is nice too read.

However, there are shortcomings of the paper:
* The proposed method does not outperform existing methods (on ATSP). We agree however that outperforming highly optimized codes like LKH is not necessarily expected.

Due to the novelty of the architecture and the tackled setting and the relevant results the paper is a good fit for NeurIPS.

We urge the authors to reflect the discussion and the additional experiments in the final paper and include the promised changes in their final paper.